# Five Guidelines for Adopting Smartwatches in Construction: A Novel Approach for Understanding Workers' Efficiency Based on Travelled Distances and Locations

Cristina Toca Pérez *, Stephanie Salling and Søren Wandahl

Department of Civil and Architectural Engineering, Aarhus University, 8000 Aarhus, Denmark;
stsa@cae.au.dk (S.S.); swa@cae.au.dk (S.W.)
* Correspondence: cristina.toca.perez@cae.au.dk

**Abstract:** This study is part of an ongoing research project aiming to develop a method for understanding workers' efficiency (workers' time spent in value-adding activities) by measuring new indexes, such as workers' travelled distances and workers' locations collected by smartwatches. To achieve the objective of the study, a Design Science Research (DSR) strategy was adopted. The first cycle consists of understanding which types of information smartwatches can collect and how this data can be employed for measuring workers' efficiency. This paper reports a case study as part of the first Cycle of the DSR. The object studied were the activities carried out by a carpenter trade in a housing renovation project. The authors used the geographic coordinates obtained by smartwatches worn by the carpenter trade connected to two Global Navigations Satellite Systems. The primary contribution of this research consists of proposing a set of five guidelines for the application of smartwatches, using data gathered from the case study. The guidelines are: (1) adopt a stratified sampling approach for selecting the workers involved according to their tasks conducted; (2) set up the smartwatches considering workers' physical features; (3) carefully consider the job site location for delivering the smartwatch to workers; (4) establish assumptions for the data cleaning process regarding construction project features and the study's goal; and (5) use individual participant data in the analysis according to each participant's characteristics and role.

**Keywords:** construction site; transportation activities; travelled distance; smartwatch

## 1. Introduction

In the last decades, significant technological and scientific advances have been made as a result of Industry 4.0, which mainly focuses on applying computers, sensors, and cyber-physical systems in production of goods or services. The construction industry has also benefited from this development, resulting in the term Construction 4.0, which has gained popularity in the recent years [1]. The new technologies offer new opportunities for companies that want to increase the quality of their work, complete projects on time, and offer new services to their customers [2]. The adoption of Industry 4.0 technologies will enable the improvement of productivity and, by extension, greater market success [3].

Among the different possibilities of Construction 4.0, one is to use sensors or trackers to measure workers' activities to understand their performance based on how and where workers spend their time on the construction sites [4]. With regards to how workers spend their time, several methods for improving construction labor producitivty and several indexes for measuring construction labor productivity, either direct or indirect, have been created and adopted within the construction industry [5]. These indexes aim to evaluate how effectively equipment and workforce utilization are managed. An often applied approach is Activity Analysis, which determine how workers spend their time on different work activities, mainly classified into Value-Added Work (VAW), and Non-Value-Added-

Work (NVAW) [5]. With regards to where workers spend their time on the construction site, previous studies adopted location-based technologies [6].

It is recognised that the first step towards developing an automated monitoring system for construction workers' performance consists of establishing a complete and competent activity recognition solution [4]. A typical data-driven and automated construction monitoring system consists of three sequential levels: (1) activity recognition, which involves the development of technology that determines which type of activity is taking place at any given time; (2) activity tracking, which exploits information from recognised activities to trace workers in different time periods and different locations; (3) performance monitoring, which aims to determine the progress of activities against planned schedules and preestablished Key Performance Indicators (KPI) [7].

The main existing technologies adopted in previous research works to automate data collection on construction workers can be grouped into two approaches: (1) computer vision-based technologies; and (2) sensor-based technologies. Vision-based activity analysis requires single or multiple cameras for detecting and tracking resources as well as procedures for activity recognition [8]. Sensor-based technologies enable the identification of measurement of workers' posture, motions, and location [9]. Among the existing digital approaches for data collection, sensor-based technologies using body-worn sensors have gained greater attention among researchers for monitoring construction activities [10] due to their flexibility to adapt to different external conditions and their reduced size easily to be embedded in e.g., wristbands [11].

Body-worn sensors, also called wearable sensors, are small devices that people can carry around while performing their daily activities [12]. The terms 'wearable technology', 'wearable devices', and 'wearables' all refer to electronic technologies or computers that are incorporated into items of clothing and accessories which can comfortably be worn on the body [13]. These wearable sensors can integrate accelerometers, gyroscopes, and magnetometers, collectively called Inertial Measurement Units (IMUs). IMUs can measure inertial body motions in three axes, as each activity creates unique acceleration signal patterns. Machine learning algorithms are commonly used to differentiate diverse activities by learning the signal pattern conditions [10,11].

Location-based sensors like Global Navigations Satellite System (GNSS), Global Position System (GPS), Radio-frequency Identification (RFID), and ultrawideband (UWB) can track workers' real-time location changes to automatically collect worker-activity-related data [5]. However, the application of sensor-based technologies often requires expensive equipment and an extended data analysis period due to the necessity of extensive training data sets for the machine learning process. Hence, these advanced tools can be challenging to implement by practitioners in construction projects. To avoid these issues, the authors chose to use smartwatches for this study. Some of the advantages of smartwatches are their low cost, user-friendly interface, and that they are comfortable and easy to wear and use.

Today's smartwatches have many integrated sensors, used extensively in current activity recognition studies [14]. Several studies agree that this kind of wristband activity tracker is an objective and non-intrusive way of continuously observing measurements that are important for the performance monitoring of construction projects [4]. This paper presents a new approach to use the geographical location data of workers collected through smartwatches as means of remote activity tracking, which complements the current body of knowledge by providing a source of evidence that can potentially increase the accuracy of activity tracking studies. The authors of this study developed a method for understanding workers' efficiency (workers' time spent in value-adding activities) indirectly by measuring new indexes, such as: workers' travelled distances and workers' locations collected by smartwatches. To achieve the objective of the study, the first step consists of understanding which type of information smartwatches can collect and how this data can be employed for measuring workers' efficiency. Hence, this study was driven by the following research questions:

- How can smartwatches be adopted to facilitate understanding workers' travelled distances and job site location?
- How can the data gathered using smartwatches be helpful in measuring workers' efficiency?

To address these questions, the authors conducted an exploratory case study. This study is part of an ongoing research project which aims to improve on-site efficiency in the Danish construction sector by improving resource productivity. The result section in this paper is mainly descriptive, as it is focusing on identifying the potential uses of the smartwatches rather than presenting a statistical analysis of the workers' travelled distances and their job site locations.

This paper is structured in seven sections, organised as follows. After this introduction, the Section 2 presents the theoretical background of the research. Firstly, this section presents, based on the existing literature, the characteristics, advantages, and limitations of adopting smartwatches as a suitable device for measuring workers' travelled distances and locations. Then, the section examines the literature on the adoption of smartwatches in the construction industry. In the Section 3, the research method is described. This section includes the research strategy justification and the construction project description where the empirical case study was carried out. Finally, this section explains the research design, including a description of the steps conducted. The results are presented and described in the Section 4. The Section 5 presents the primary contribution of this study consisting of a set of guidelines for further application of smartwatches to collect workers' travelled distances and locations. The Section 6 discusses other possible contributions of the present study. Lastly, Section 7 introduces the conclusions obtained from the study performed and lists the actions that will be carried out to continue the initiated work.

## 2. Literature Review

### 2.1. Smartwatches

Smartwatches, smartphones, and other mobile devices have become popular in supporting human activity recognition for research purposes, in contrast with other complex devices that require sophisticated laboratories and have minimal mobility [15]. Smartwatches are convenient to wear and have the capability to collect data in a continuous manner, given that the battery is charged periodically [16]. Smartwatches have already been widely used and accepted among the general population. There are applications taking advantage of these devices, based on the analytics of their captured data to provide personalised services to users [17]. One of the most popular applications is electronic fitness trainers, which allow the users to generate training plans based on their physical conditions and goals. This type of application has received good acceptance in the consumer market, and have promoted the quick popularisation of smartwatches [17].

There are several smartwatch brands in the market, some of the most popular being Garmin®, Polar®, Suunto®, and TomTom® [18]. Within these brands, there are various existing models, e.g., Garmin Forerunner® 45, which is the model used in this research study. Most smartwatches give information about distance and speed using a GNSS, such as GPS. To obtain the best possible GPS readings, a high sampling frequency, open areas free from obstructions such as tall buildings, and clear skies are required [19]. Johansson et al. [18] conducted a case study to determine the accuracy of various GPS sport watches in measuring distance throughout a 56 km running race. They concluded that the GPS sport watches in the study have an accuracy of $0.6 \pm 0.3\%$ to $1.9 \pm 1.5\%$ (median $\pm$ interquartile range) in reporting distance covered. This indicates that GPS sport watches are a valid and feasible method for sport scientists and coaches to measure performance and track training load. However, the accuracy could differ depending on the field of application. To the authors' knowledge, the accuracy of smartwatches has not yet been tested on construction sites, where several solid objects (e.g., concrete structures) could potentially interfere with the GNSS signal. Hence, the aspect of accuracy needs to be considered when data are interpreted in other fields.

Despite the advantages of using smartwatches in terms of gathering data, the transfer process of the collected data to other devices presents some limitations [17]. Generally, there are two main approaches: (1) access to raw data through a smartphone; and (2) access to data through an app provided by the wearable proprietary warehouse. The first option enables the researcher to perform real-time studies. However, to transfer the data from the wearable to the smartphone, a specific app needs to be developed, and this requires programming knowledge about e.g., event communication. Moreover, depending on the brand of the wearable, an app in the wearable itself also needs to be developed. In the second option, data is transferred through proprietary Application Programming Interface (APIs) based on a Representational State Transfer (REST) service provided by the wearable provider. The main drawback of this approach is that real-time access to data is not possible, because it depends on the continuous synchronisation between the wearable and the warehouse [17]. In the construction industry, during the construction phase, the visualisation of real-time worker locations could benefit safety engineers and managers to react in real-time to an incident. However, for the purpose of this study, not having access to real-time data from the smartwatches is not considered a critical limitation. The authors of this paper consider it sufficient to obtain the data after synchronising at the end of each workday.

Probably, the most significant limitation of the adoption of smartwatches is that they collect location from a GNSS, as mentioned above. A GNSS receives signals from more than three satellites. With these, it can accurately estimate the current location through trilateration by the rear intersection, using three different distances [20]. The accuracy of GPS, one of several GNSS satellite systems, is within 4.9 m (16 ft.) radius under open sky [21]. However, in an indoor environment, GNSS cannot be used. The GNSS signals are carried through waves, which have a frequency that does not move easily through solid objects, such as walls of buildings. [20]. For this reason, the approach presented in the present paper is limited to construction projects where workers spend most of their working time outdoors (e.g., renovation building projects with predominantly external activities and civil infrastructure projects such as roads and bridges).

### 2.2. Previous Studies That Used Wristworn Sensors in the Construction Sector

Previous studies that adopted wristworn sensors in the construction industry mainly aimed to automate the worker's Activity Recognition (AR) or monitor workers' Health and Safety (HS) conditions. Table 1 presents 19 papers found in the existing literature that adopted these devices. Five of them adopted this wearable technology for AR purposes; ten of them focused on HS management purposes; and four papers conducted Desk-Study (DS) approaches.

For workers' activity recognition, studies mainly focus on associated IMUs to the movements and motions executed tied to performing a specific construction task. The first research work published that adopted a smartwatch, specifically the GearLive from Samsung, for workers' activity recognition was published in 2012 by Cezar [22]. The author collected the data from a 3D accelerometer and 3D gyrometer from three workers on three different days and labelled them during each activity. Cezar [22] evaluates five machine learning algorithms to classify four construction activities (hammering, sawing, sweeping, and drilling). The results showed the algorithms allow for classification of activities with an accuracy of 91%. Ryu et al. [23] conducted a similar study for action recognition using a wristband-type activity tracker, specifically the eZ430-Chronos sport watch. The authors proposed and tested an approach based on accelerometer-based action recognition in five workers of a masonry trade. Innovative research was recently conducted in the activity recognition field by Jassmi et al. [4]. The authors used physiological signals as an additional source of information that helps improve the accuracy of machine learning classifiers to recognise construction labour activities. For this, they combined the use of a wristband biosensor unit with a heart rate monitoring chest strap on workers during their working hours.

Regarding the use of smartwatches to collect physiological signals for health and safety management, most of the papers found in the literature are part of a long-term research project conducted by researchers from the University of Michigan [24–27] and Pennsylvania State University [28]. In all these studies, the research group adopted the Empatica E4 to collect physiological signals, which is a wristband biosensor especially designed for researchers and physicians who are conducting research on physiology in daily life and is not for consumer use [29].

Hwang et al. [30] conducted one of the first studies in this field for understanding the challenges and opportunities to measure workers' physical demands through field Energy Expenditure (EE) measurements. Hwang et al. [30] used workers' Heart Rate (HR) to estimate EE. Continuing the interest to estimate EE using smartwatches, Jebelli et al. [26,27] collected information from 10 construction workers on a job site to examine their physiological signals when executing different construction tasks. Jebelli et al. [26,27] used the EE to separate tasks into low-, moderate-, and high-intensity activities. Jebelli's studies contributed to the body of knowledge on the in-depth understanding of construction workers' stress on construction sites by developing a non-invasive means for continuous monitoring and assessing workers' stress. Prior to this, Jebelli et al. [25] confirmed the feasibility of the wristband-type wearable sensor to evaluate construction workers' physical and mental state. To do this, Jebelli et al. [25] investigated three bio signals of two workers for 10 h to detect their physical and mental conditions during their work on the site. The three bio signals were: (1) Electrodermal (EDA), measuring the changes in the electrical properties of the skin; (2) Skin Temperature (ST), measuring body temperature; and (3) Photoplethysmogram (PPG), measuring the blood-volume variations in vascular tissues. Another study conducted by the group [24] focused on investigating the bio signal EDA to understand construction workers' perceived risk during their ongoing work. To achieve this objective, the authors [24] collected 30 h of physiological sensory data from eight construction workers during their ongoing work. The main contribution of this study was to show the flexibility of using wearable sensors to understand workers' perceived risk in construction sites continuously.

The last study published by the referred group of researchers from the University of Michigan, conducted by Shakerian et al. [28], aimed to assess the occupational risk of heat stress in construction. For this purpose, the authors collected physiological signals from 18 workers while performing specific construction tasks under three predetermined environmental conditions with a different probability of exposure to heat stress. The analysis results revealed that the proposed process could predict the risk of heat strain with more than 92% accuracy.

Still in the adoption of smartwatches for health and safety management, Guo et al. [31] used a Basis Peak$^{TM}$ smartwatch to collect workers' psychological status to associate with unsafe behaviour in order to prevent accidents. To do this, the authors ran an experiment for 18 days on workers conducting activities in a high-rise building job site. The results showed significant correlations between workers' psychological status and physical status.

The four desk-research studies identified focused on both health and safety management and how smartwatches and other wearable technologies have been used in previous studies [7,32,33], and on the construction workers' possible acceptance of the adoption of these technologies [13]. The literature review concerning applications of wearable kinematic and physiological sensors in construction safety and health conducted by Ahn et al. [32] revealed five general applications: (1) preventing musculoskeletal disorders, (2) preventing falls, (3) assessing physical workload and fatigue, (4) evaluating hazard-recognition abilities, and (5) monitoring workers' mental status. Awolusi et al. [33] concluded from their review that a wide variety of wearable technologies are being used in other industries to enhance safety and productivity while few applications are observed in the construction industry. The study conducted by Choi et al. [13] aimed to investigate determinants for workers' adoption of wearable technology in the occupational work context. To achieve this, the authors developed and applied a survey questionnaire to 120 workers at three

construction sites in the United States of America. The research results indicated that perceived usefulness, social influence, and perceived privacy risks are associated with workers' intention to adopt smart-vests and smartwatches.

The present literature review indicated growing attention to wristband-type activity trackers, such as smartwatches, as a tool to monitor and track construction workers' activities. The existing literature emphasises a high accuracy in their results for activity recognition based on the data gathered by the biosensors, accelerometers and gyroscopes embedded in the smartwatches. However, a significant limitation of those approaches consists in the data labelling process. Researchers need to select and classify a limited number of activities. Hence, this sophisticated approach can be very impractical for adoption by practitioners. Other functionalities presented in those smartwatches related to their location-based sensors were, to the author's knowledge, not yet tested. So, the location information collected by the GNSS receiver of these devices can be helpful in understanding workers' movements within the job site. To bridge this gap, the present exploratory study aims to adopt smartwatches to determine the distance travelled by workers and workers' locations during their working hours.

Table 1. Previous studies that adopted wristworn sensors in the construction industry. AR: activity recognition, HS: health and safety management, DS: desk-study.

| # | Ref. | Field | Index Measured | Topic Evaluated | Sensors/ Tools Used | Goal | Type of Wearable | Testing Environment |
|---|------|-------|----------------|-----------------|---------------------|------|------------------|---------------------|
| 1 | [22] | AR | Heart rate (HR) | Four activities (hammering, sawing, sweeping, and drilling) | Accelerometer Gyrometer | To classify construction activities using accelerometer and gyrometer data | Samsung GearLive Smartwatch | Outdoor Job site |
| 2 | [34] | HS | HR | Workers' physical strain | HR sensor ECG sensor | To monitor workers' physical strain based on HR | Basis PeakTM fitness tracker | Job site |
| 3 | [30] | HS | HR | Energy Expenditure (EE) | Accelerometer | To estimate energy expenditure (EE) by monitoring HR | Basis PeakTM fitness tracker | Job site |
| 4 | [23] | AR | HR | Four activities (spreading mortar, bring and laying blocks, adjusting blocks, and removing remaining mortar) | Accelerometer | To recognise masonry actions | eZ430-Chronos sport watch | Indoor Masonry Training Centre |
| 5 | [35] | AR | Heart rate reserve (HRR) | Work patterns | HR sensor | To investigate the relationship between work patterns and heart rate reserve measurements | "off-the-shelf"-type | Job site |
| 6 | [13] | DS | Intention to adopt | Usefulness (PU), Social influence (SI), Perceived privacy risk (PR) | Questionnaire | To investigate workers' intention to adopt a smart vest and a wristband | Basis PeakTM fitness tracker | Job site |
| 7 | [31] | HS | HR Number of steps | Level of stress: (1) Calm; (2) Stress; and (3) Sport status | HR sensor | To investigates the relationship between workers' psychological status and physical data | Basis PeakTM smartwatch | Job site |
| 8 | [25] | HS | Electrodermal (EDA), Skin Temperature (ST), Photoplethysmogram (PPG) | Workers' physical and mental state | Biosensor | To assess workers' physical and mental state based on three bio signals. | Wristband sensor | Job site |
| 9 | [33] | DS | - | Predicting safety performance | Literature Review | To review the applications of wearable technology for personalised construction safety monitoring | - | - |
| 10 | [10] | AR | Hand activities | Four masonry activities (spreading mortar, bringing and laying blocks, adjustments blocks, and removing remaining mortar) | Accelerometer | To recognise actions of masonry work for automatic field data collection | eZ430-Chronos sport watch | Laboratory |

**Table 1.** *Cont.*

| # | Ref. | Field | Index Measured | Topic Evaluated | Sensors/ Tools Used | Goal | Type of Wearable | Testing Environment |
|---|------|-------|----------------|-----------------|---------------------|------|------------------|---------------------|
| 11 | [26] | HS | HR | Level of stress: (1) low; (2) medium, and (3) high | Biosensor | To measure workers' stress with signals from a wristworn biosensor | Empatica E4 wristband biosensor | Job site |
| 12 | [27] | HS | HR | Physical demand: (1) low; (2) moderate, and (3) high | Biosensor | To measure workers' physical demand levels from a wristworn biosensor | Empatica E4 wristband biosensor | Job site |
| 13 | [24] | HS | EDA | Workers' perceived risk: (1) low; and (2) high | Biosensor | To understand workers' perceived risk by monitoring physiological responses | Empatica E4 wristband biosensor | Job site |
| 14 | [32] | DS | - | Five applications to evaluate and prevent: (1) musculoskeletal disorders, (2) falls, (3) physical workload, (4) hazard-recognition abilities, and (5) workers' mental status | Literature Review | State-of-the-art within applications of wearable kinematic and physiological sensors in construction safety and health | - | - |
| 15 | [7] | DS | - | Three categories: (1) audio-based, (2) kinematic-based, and (3) computer vision-based techniques | Literature Review | State-of-the-art of literature concerning technologies for automated performance monitoring of construction workers and equipment | - | - |
| 16 | [36] | HS | HR Sleep quality | Fatigue | Biosensor | To estimate fatigue based on monitoring HR and sleep quality of construction workers | Fitbit Charge 2 | Laboratory |
| 17 | [28] | HS | ST, PPG, EDA | Heat-related illnesses | Biosensor | To propose a predictive heat strain process based on workers' physiological signals to address the risk of heat-related illnesses | Empatica E4 wristband biosensor | Laboratory |
| 18 | [37] | HS | HR | Workers' perceived risk: (1) low; and (2) high | Biosensor | To develop an automatic method for identifying workers' perceived risk level using a wristworn biosensor collecting physiological signals | Empatica E4 wristband biosensor | Job site |
| 19 | [4] | AR | Blood volume pulse (BVP), Respiration rate (RR), HR, Galvanic skin response (GSR), ST | Four activities (Oiling, moving, cleaning and opening stone moulds) | Biosensor Accelerometer | To recognise actions using labour physiological data | Empatica E4 wristband biosensor | Pre-fabrication factory |

## 3. Research Methodology

### 3.1. Research Strategy

This research project adopted a Design Science Research (DSR) strategy, which aims to produce innovative constructions, called artifacts, to solve real-world problems and to contribute to the theory of the discipline in which it is applied [38]. This project's artifact consists of a method for understanding workers' efficiency indirectly by measuring workers' travelled distances and workers' locations collected by smartwatches worn by construction site workers. The research process of designing the artifact developed involved four learning cycles, named Cycle 1, 2, 3, and 4. During each cycle, the four main phases that categorize this DSR were conducted: (1) understanding; (2) construction; (3) analysis; and (4) evaluation.

The four DSR phases organize different approaches carried out by the research team in various construction sites, named Case A, B, C and D. For that purpose, this research used the case study [39] method as the primary research strategy, as case studies offer flexibility for explorative and theory-building research in real-life contexts. During a case study, the research scope can be re-addressed and complementary data sources can be acquired, and the method serves several research objectives [40].

The present paper exclusively focuses on presenting the results of the first learning cycle conducted in Case A. Among the number of options in carrying out case research, the authors characterise the first study as an exploratory case study [39]. The authors adopted the exploratory case study method because it enables the investigation of a given phenomenon characterised by a lack of detailed preliminary research. The phenomenon of the study comprised construction workers' travelled distances and workers' locations using smartwatches as a digital tool for collecting data. The lack of previous studies that used smartwatches for this purpose marks this study as exploratory. The real-life context is represented by the building project studied.

### 3.2. Case Study Description

The study was conducted on a building renovation project located in the city of Odense in Denmark, named in this research work as Case A. The housing complex consists of two- to four-story buildings with two apartments on each floor. There are a total of 587 housing units. The buildings were first established in the early 1950s and were, at the time of this research, undergoing comprehensive renovation. The renovation included replacing old balconies, windows, kitchens, and bathroom interiors, adding insulation in walls, putting up drywall partitioning walls, and turning some units into accessible housing units by installing elevators in the stairwells. During the execution of the renovation project, tenants were rehoused in the period when their apartment was being renovated, but they were living in their apartment during the renovation of the neighbouring buildings. To minimise the need for rehousing, only around 15% of the units were renovated at the same time. The result of this agreement was that the contractor had a restricted area for the construction work activities. For this reason, logistical challenges were significant in the project.

This project was chosen for two reasons. First, the possibility of utilising an existing collaboration with the construction company for the development of this research. The partnership allowed the researchers to access project-related documents, observation of routine activities, as well as interact with team members during the data collection period on the job site. Second, this type of building project represents the typical Danish social housing buildings that have been or will be retrofitted in the upcoming years as part of the Danish government's strategy for energy renovation of the existing building stock [41,42]. Analysing a popular construction solution in a typical social housing building will allow the authors to generalise case learnings to similar contexts about how and where workers spend their working hours in renovation projects. Thus, the final goal of this exploratory case study is the creation of hypotheses for further analysis [39] rather than quantitatively stating statistical facts.

The construction site layout can be seen in Figure 1. The site consisted of inhabited buildings, buildings under renovation (Blocks 51, 53, 55, 57, 58, 59, 60, 62, 64, 66, and 68 in Figure 1), and two large warehouses for material storage. The rest of the job site was allocated for other support activities, such as transport space, temporary storage space, and hazard space. Vehicles, pieces of equipment, and workers shared the same paths for transport activities. Tools and small pieces of equipment were stored in modular storage containers along with the construction site. Office containers, break rooms, and changing rooms were placed in a two-story complex located on the perimeter of the site.

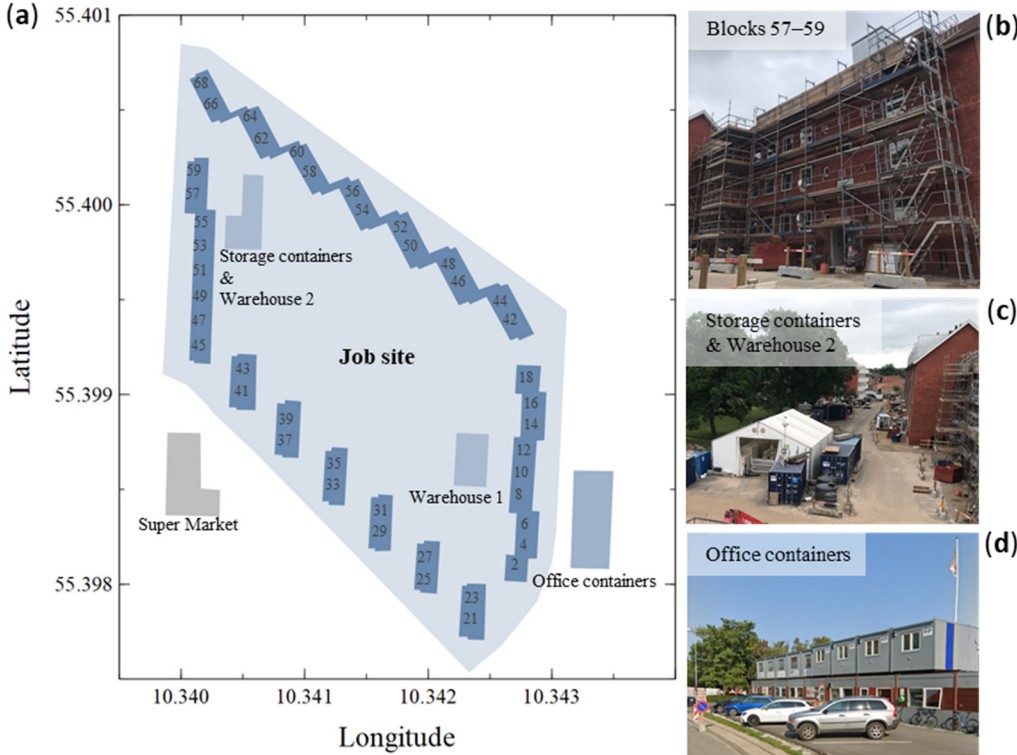

**Figure 1.** (**a**) Job site layout including geographic coordinates; (**b**) Blocks 57–59; (**c**) Storage containers and Warehouse 2; and (**d**) Office containers.

### 3.3. Research Design

The research comprised six steps: (1) study preparation; (2) setup of smartwatches; (3) gathering data; (4) data storing and transformation; (5) data cleaning; and (6) data analysis.

### 3.3.1. Study Preparation

The first stage of this exploratory study aimed to define the project goal and to select the digital devices for collecting data. For that purpose, a meeting with field engineers and managers was conducted at the job site in the beginning of June 2021, week 23. During this meeting, the identification of the period of job site visits and the trade involved in the study were defined.

The Garmin Forerunner 45 smartwatch (see Figure 2a) was chosen for this study due to its technical features, such as: (1) the availability to use without a phone; (2) the availability to transfer data from the smartwatch to the laptop using the Garmin Express app, and then access historical activity through the Garmin Connect website; (3) the availability to export data in multiple formats including the GPS Exchange Format (GPX), which allowed the authors to transform the raw data from the website to a Comma-Separated Value (CSV) format; and (4) the availability to create and customize activities, which allowed the authors to manually enter an activity in accordance with the characteristics of the relevant process using basic information.

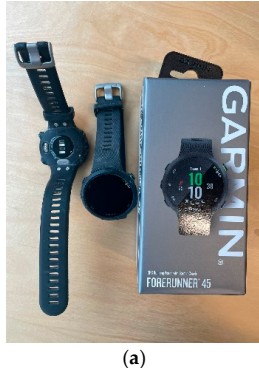 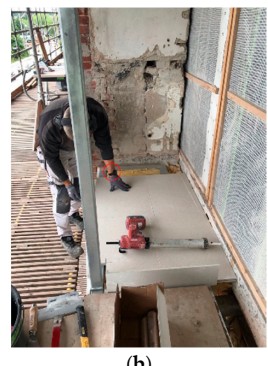 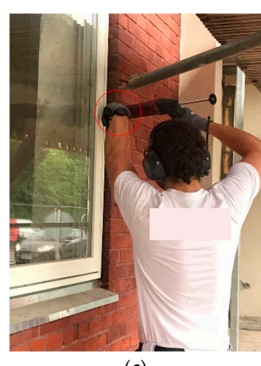

| (**a**) | (**b**) | (**c**) |

**Figure 2.** (**a**) Garmin Forerunner 45 adopted for the study; (**b**,**c**) two examples of carpenters wearing the smartwatch.

Prior to the data collection period, the participants of the study were identified. The carpenter trade was chosen as the subject of study because this trade is part of the same company as the general contractor, and the general contractor was most interested in understanding the time utilization of its own trade. Carpenter tasks include outdoor work such as installing new balconies and windows (see Figure 2b,c), and indoor work in the form of installing new partition walls, baseboards, and doors, and insulation of floors and roofs. The number of study participants was limited to ten because of the maximum number of available smartwatches. Hence, the participant sample for the study represents 45% of the population size of the carpenters' trade (22 workers). The ten carpenters were chosen randomly, independent of which task they were assigned to. All participants gave their informed consent for inclusion before they participated in the study.

### 3.3.2. Setup of Smartwatches

The preparation of the smartwatches involved several aspects: (1) creating a user account for each smartwatch; (2) creating a group of all user accounts; (3) setting up preferences (e.g., notification settings, units of measure, GPS activation); and (4) adding user information (height, weight, age). Personal data policies hindered the collection of personal user information in this study. Consequently, general, average data for a Danish male [43] was used as a substitute for the specific numbers. Table 2 summarises the parameters adopted for the study.

**Table 2.** Parameters adopted for the study.

| | Sounds & Alerts | Activity Type | Activity Tracking | | | User Settings | | | |
| --- | --- | --- | --- | --- | --- | --- | --- | --- | --- |
| | | | Activity Tracking | HR | Measurement Units | Height | Weight | Gender | Age |
| **Setting** | All off | Walking | On | Wrist | Metric | 182 cm | 86 kg | Male | 41 |

### 3.3.3. Gathering Data

For gathering data at the construction site, ten weekdays of on-site data collection (eight hours/day) during weeks 25 and 26 of 2021 were conducted. During this period, the authors were able to observe all the activities involved in the renovation process of a housing unit. Hence, the sample size of ten days made it possible to look for patterns in the data and come up with a model to view this data. The first day of the data collection period, named Day 0, was used to identify the carpenters' tasks, and get familiar with using the smartwatches. The subsequent nine days were used to collect data. Those days were named Day 1 to Day 9. Day 0 was Monday of week 25 and Day 9 was Friday of week 26. The data collection duration was the same as the workers' workhours from 06:30 to 14:30, excluding a coffee break (09:00 to 09:15) and a lunch break (10:30 to 11:00). On Fridays (Day 4 and Day 9 in this study), the workday was one hour shorter, i.e., 07:00 to 14:00.

The data collection aimed to gather two kinds of data: (1) quantitative data regarding the workers' travelled distances and locations; and (2) qualitative data regarding workers' perception of wearing a smartwatch while performing their work.

Smartwatches Delivered to Workers

The smartwatches were handed out to the carpenters each morning at the beginning of the workday. The smartwatch activity (called "Walking" on the watch) was started by one of the researchers before handing out the watch. The delivery mainly took place by a material storage area close to warehouse 2 (see Figure 1). Some watches were also handed out by the office containers (see Figure 1) because some carpenters went to the break room to change clothes before starting their workday and then approached the researchers to pick up a watch. At the end of the workday, the smartwatches were collected by the researchers either at the locker room or at warehouse 1. Then, the researchers ended and saved the smartwatch activity.

Workers' Perception

On the last day of the job site visits, the authors collected feedback from eight of the ten participants through a questionnaire. The remaining two participants did not wish to answer the questionnaire. The questionnaire aimed to collect the carpenters' perception of wearing a smartwatch during their workday. It consisted of four questions using the Likert scale and was carried out using a tablet as Computer-Assisted Personal Interviewing (CAPI). The tablet allowed to record answers directly into the survey platform and ensured the anonymity of participants.

### 3.3.4. Data Storing and Transformation

The data was acquired by connecting and synchronising the Garmin smartwatch to the Garmin Express laptop app using a USB cable. To prevent loss of data due to empty batteries or technical errors, the smartwatches were charged every night and synchronized after each day of data collection.

The files with activity recordings from the smartwatches were organised into folders named "Day 1" to "Day 9". Each folder contained ten subfolders with the data from each smartwatch, resulting in a total of 90 possible activity recordings. Unfortunately, despite the above-mentioned precautionary measures, during the data collection, different issues prevented data from being collected for 26 of the activities. In 18 cases, the problem was the absence of the participating carpenters due to, e.g., illness or education leave. The remaining eight unusable activities were either due to the activity accidentally being ended by the carpenter before the workday ended, not being saved correctly, or the wrong kind of activity being used (for instance, using "Indoor walking" as activity would mean GPS coordinates were not recorded).

Table 3 summarises the file status from the smartwatches, using the following legend: (1) Yes represents the files that were saved correctly; (2) No corresponds to smartwatches that were not used during that day due to absence of workers involved in the study; (3) Not Saved (NS) corresponds to files, which were not usable for any of the reasons mentioned above. As shown in Table 3, the days with the lowest number of activities saved were Day 1 and Day 9, with five files each day. Day 5 represents the day with most files saved, namely nine files. As the smartwatches were randomly distributed among the carpenters each day, there is no pattern in which watches were used on which days. A total of 64 usable activities with GPS coordinates were recorded, which gives an average of seven watches from each day of data collection.

Hence, on each of the nine days of data collection, between five and nine of the ten smartwatches were in use. To organise this data, the authors generated a single XLSX-file including the data from the 64 usable files.

**Table 3.** Status of files from smartwatches.

|  | SW01 | SW02 | SW03 | SW04 | SW05 | SW06 | SW07 | SW08 | SW09 | SW10 | Total |
|---|---|---|---|---|---|---|---|---|---|---|---|
| **Day 1** | No | Yes | Yes | Yes | Yes | NS | No | No | No | Yes | 5 |
| **Day 2** | Yes | Yes | Yes | Yes | NS | No | Yes | Yes | Yes | NS | 7 |
| **Day 3** | Yes | Yes | NS | Yes | No | No | No | Yes | Yes | Yes | 6 |
| **Day 4** | Yes | Yes | Yes | Yes | No | Yes | No | Yes | Yes | Yes | 8 |
| **Day 5** | Yes | Yes | Yes | Yes | Yes | Yes | Yes | Yes | No | Yes | 9 |
| **Day 6** | No | Yes | NS | Yes | Yes | Yes | Yes | Yes | Yes | Yes | 8 |
| **Day 7** | NS | Yes | No | Yes | Yes | Yes | Yes | Yes | Yes | Yes | 8 |
| **Day 8** | No | Yes | Yes | Yes | Yes | No | Yes | Yes | Yes | Yes | 8 |
| **Day 9** | No | NS | No | Yes | Yes | No | NS | Yes | Yes | Yes | 5 |
| **Total** | 4 | 8 | 5 | 9 | 6 | 4 | 5 | 8 | 7 | 8 | 64 |

As previously mentioned, the data came from the Garmin website in a GPX-format. It was transformed to a CSV-format using Python programming. Python is a general-purpose programming language [44]. The features identified for each data point are: (1) Unix time, consisting of the number of seconds that have elapsed since the Unix Epoch (1 January 1970); (2) the time, day, and hour (3) latitude (4) longitude; (5) altitude; and (6) the accumulative distance; These features were organised in a Microsoft Excel Worksheet (XLSX).

### 3.3.5. Data Aggregation and Cleaning

The data was aggregated and expressed in a summary according to the smartwatches used (e.g., SW01) and the day of the data collection (e.g., Day01 or D1) for further analysis, see Table 4. The data was cleaned considering three assumptions to improve the validity and achieve the required accuracy of the smartwatch data according to the purpose of this study.

First, to ensure comparability, all activities lacking data from more than one continuous hour of the workday were excluded from the study. Hence, the activity duration of each activity saved considered valid for the analysis was 8 h $\pm$ 30 min for all days, except for Fridays (Day 5 and Day 9) where it was 7 h $\pm$ 30 min. The excluded activities are highlighted in Table 4. Excluding these ten activities resulted in a total of 54 activities to be analysed, totalizing 194,549 data points. That is, around 3600 data points per smartwatch.

Second, all data collected during the lunch break (i.e., 10:30–11:00 every day) was removed, since the breaks are not part of the paid worktime. For the same reason, data collected outside the daily workhours was removed as well.

Third, the data was cleaned according to the speed of walking. If the speed from travelling from two consecutive coordinates was lower than 0.5 m/s it was assumed the worker was standing, thus this should not be considered a travelled distance. If the speed exceeded 1.48 m/s the worker was considered to be running. Data points indicating a speed much higher than 1.48 m/s could also be caused by GPS errors [19], e.g., due to work tasks being carried out inside a building. Since the data of interest only considered movement at walking speed, data points below and above the described limits were removed. The data cleaning according to speed reduced the size of the stored data to approximately one third, resulting in a reduction from 194,549 to 63,145 data points.

### 3.3.6. Data Analysis

The analysis of data collected and cleaned (see Table 5) aimed to identify and analyse the distance travelled by the carpenters and their location on the job site. For interpreting the distance, the authors studied three kinds of results: (1) total travelled distance; (2) cumulative travelled distance; and (3) average travelled distance. The analysis of the indexes focused on discussing how this information could be useful for the existing production planning systems. The data of each smartwatch was binned into 30-min intervals to facilitate comparison among all days. Using an interval size of 30 min made it possible to exclude the lunch break from the analyses.

**Table 4.** Duration of saved activities [hh:mm:ss].

| Device Name | Activity Duration | Starting Time | Ending Time | Device Name | Activity Duration | Starting Time | Ending Time |
|---|---|---|---|---|---|---|---|
| D1-SW02 | 08:01:07 | 06:38:06 | 14:39:13 | D6-SW02 | 07:38:01 | 06:37:06 | 14:15:07 |
| D1-SW03 | 07:53:01 | 06:36:30 | 14:29:32 | D6-SW04 | 07:46:01 | 06:38:40 | 14:24:41 |
| D1-SW04 | 08:00:44 | 06:27:25 | 14:28:09 | **D6-SW05** | **07:21:57** | **06:36:23** | **13:58:20** |
| D1-SW05 | 07:48:51 | 06:44:57 | 14:33:48 | D6-SW06 | 07:48:04 | 06:35:07 | 14:23:11 |
| D1-SW10 | 07:57:09 | 06:32:14 | 14:29:23 | D6-SW07 | 07:54:38 | 06:35:03 | 14:29:41 |
| D2-SW01 | 07:44:44 | 06:37:00 | 14:21:44 | D6-SW08 | 07:49:07 | 06:35:36 | 14:24:43 |
| D2-SW02 | 07:54:16 | 06:33:44 | 14:28:00 | **D6-SW09** | **05:05:22** | **06:36:51** | **11:42:13** |
| D2-SW03 | 07:44:26 | 06:31:29 | 14:15:55 | D6-SW10 | 07:54:37 | 06:35:18 | 14:29:55 |
| D2-SW04 | 07:50:07 | 06:33:20 | 14:23:27 | D7-SW02 | 07:57:35 | 06:29:39 | 14:27:14 |
| D2-SW07 | 07:55:37 | 06:31:41 | 14:27:18 | D7-SW04 | 07:52:20 | 06:30:06 | 14:22:26 |
| D2-SW08 | 08:00:23 | 06:27:18 | 14:27:41 | D7-SW05 | 07:37:31 | 06:35:16 | 14:12:47 |
| D2-SW09 | 07:49:57 | 06:31:53 | 14:21:50 | D7-SW06 | 07:33:31 | 06:35:41 | 14:09:12 |
| **D3-SW01** | **07:02:06** | **07:19:45** | **14:21:51** | D7-SW07 | 07:32:51 | 06:35:28 | 14:08:19 |
| D3-SW02 | 07:51:41 | 06:33:07 | 14:24:48 | D7-SW08 | 07:49:33 | 06:29:27 | 14:19:00 |
| D3-SW04 | 07:49:02 | 06:32:40 | 14:21:42 | D7-SW09 | 07:45:46 | 06:35:27 | 14:21:13 |
| D3-SW08 | 07:50:52 | 06:31:25 | 14:22:17 | D7-SW10 | 07:56:09 | 06:33:10 | 14:29:19 |
| D3-SW09 | 07:49:43 | 06:31:50 | 14:21:33 | D8-SW02 | 07:49:01 | 06:34:37 | 14:23:38 |
| **D3-SW10** | **06:56:01** | **06:32:52** | **13:28:53** | D8-SW03 | 07:47:59 | 06:36:01 | 14:24:00 |
| D4-SW01 | 06:41:21 | 07:09:48 | 13:51:09 | D8-SW04 | 07:55:24 | 06:30:45 | 14:26:09 |
| D4-SW02 | 06:59:07 | 07:00:20 | 13:59:27 | **D8-SW05** | **07:17:38** | **07:05:49** | **14:23:27** |
| D4-SW03 | 06:38:58 | 07:15:49 | 13:54:47 | D8-SW07 | 07:47:51 | 06:40:12 | 14:28:03 |
| **D4-SW04** | **03:44:52** | **07:07:40** | **10:52:32** | D8-SW08 | 07:39:13 | 06:34:37 | 14:13:50 |
| **D4-SW06** | **06:21:34** | **07:33:21** | **13:54:55** | D8-SW09 | 07:45:17 | 06:34:45 | 14:20:02 |
| D4-SW08 | 06:49:59 | 07:14:18 | 14:04:17 | D8-SW10 | 07:45:13 | 06:34:53 | 14:20:06 |
| D4-SW09 | 06:57:29 | 07:00:09 | 13:57:38 | D9-SW04 | 06:48:53 | 07:08:57 | 13:57:50 |
| **D4-SW10** | **06:59:12** | **07:00:20** | **13:59:32** | D9-SW05 | 06:57:13 | 07:08:30 | 14:05:43 |
| D5-SW01 | 07:49:11 | 06:34:39 | 14:23:50 | **D9-SW08** | **06:29:32** | **07:27:46** | **13:57:18** |
| **D5-SW02** | **07:27:49** | **06:37:53** | **14:05:42** | D9-SW09 | 06:44:14 | 07:06:06 | 13:50:20 |
| D5-SW03 | 07:47:44 | 06:37:10 | 14:24:54 | D9-SW10 | 06:45:42 | 07:04:32 | 13:50:14 |
| D5-SW04 | 07:47:25 | 06:38:19 | 14:25:44 | | | | |
| D5-SW05 | 07:52:56 | 06:34:00 | 14:26:56 | | | | |
| D5-SW06 | 07:52:38 | 06:33:45 | 14:26:23 | | | | |
| D5-SW07 | 07:52:23 | 06:33:42 | 14:26:05 | | | | |
| D5-SW08 | 07:50:21 | 06:33:47 | 14:24:08 | | | | |
| D5-SW10 | 07:49:14 | 06:34:47 | 14:24:01 | | | | |

**Table 5.** Summary of the collected data used for analysis, after cleaning.

| Participants | Activities Analysed | Data Size of Csv Files | Total Duration [h] | Sample Size | Variables of Each Data Point |
|---|---|---|---|---|---|
| 10 | 54 | 19.6 MB | 413.5 h | 63,145 points | 2 Identifiers (ID, timestamp) 3 Location (Latitude & longitude & altitude) 1 Distance (Accumulative distance) |

To analyse workers' locations, the authors examined the distribution of data points and their density within the job site. For this, the data extracted from the smartwatches was visualised using the Veusz program, which is a scientific plotting and graphing program [45]. This allowed the researchers to plot the data using a graphical 2D user interface. The authors collected the coordinates of the job site facilities and buildings using the smartwatch and converted them to a visual layout using the RouteConverter program. RouteConverter is a GPS tool used to display, edit, and convert routes from several different file formats [46]. The list of data coordinates obtained from the RouteConverter was exported into a Microsoft Excel Open XML Spreadsheet (XLSX), converted into a CSV format, and then imported to Veusz. Hence, Veusz allowed visualising the location of each data point and the job site layout.

Still in this phase, the authors analysed the results of the questionnaire applied to workers about their perception of using smartwatches. The main contribution of this paper, however, is the presentation of a list of lessons learned from the adoption of smartwatches for understanding how and where workers spend their worktime. As a part of this, the authors analyse how the data cleaning and data analysis conducted in this study could have influenced the results obtained and explain each lesson learned.

## 4. Findings

The results of using smartwatches for tracking workers' walked distances are presented in this section using three types of analysis conducted with the data gathered during the present exploratory case study: (1) workers' travelled distance; (2) workers' location on the job site; and (3) workers' perception of wearing a smartwatch.

### 4.1. Workers' Travelled Distance

4.1.1. Total Travelled Distance (TTD) along the Day

The travelled distances logged each day of the data collection are summarised in Figure 3. Each column in the diagrams represents a smartwatch, and the data are binned per 30 min of the workday from 06:30–14:30. The line shows the average distance considering all watches in use on that day. Figure 3 makes it possible to visually compare the data from the different days. The lines fluctuate to a varying degree; however, there is a general pattern of movement throughout the workday; the lowest average distances are 30 min after the beginning of the day and at the end of the day, and the highest averages are in the first 30 min of the day and after the lunch break, which was at 10:30–11:00 (excluded from the graphs). The distances vary from 0.00 km (Day 1, SW04 at 12:30–13:00; Day 3, SW02 at 08:00–08:30) to 0.94 km (Day 5, SW06 at 07:30–08:00; Day 9, SW09 at 09:00–09:30) walked in 30 min. The mean varies from 0.12 km (Day 6, at 07:30–08:00) to 0.66 km (Day 9, at 11:00–11:30) walked in 30 min.

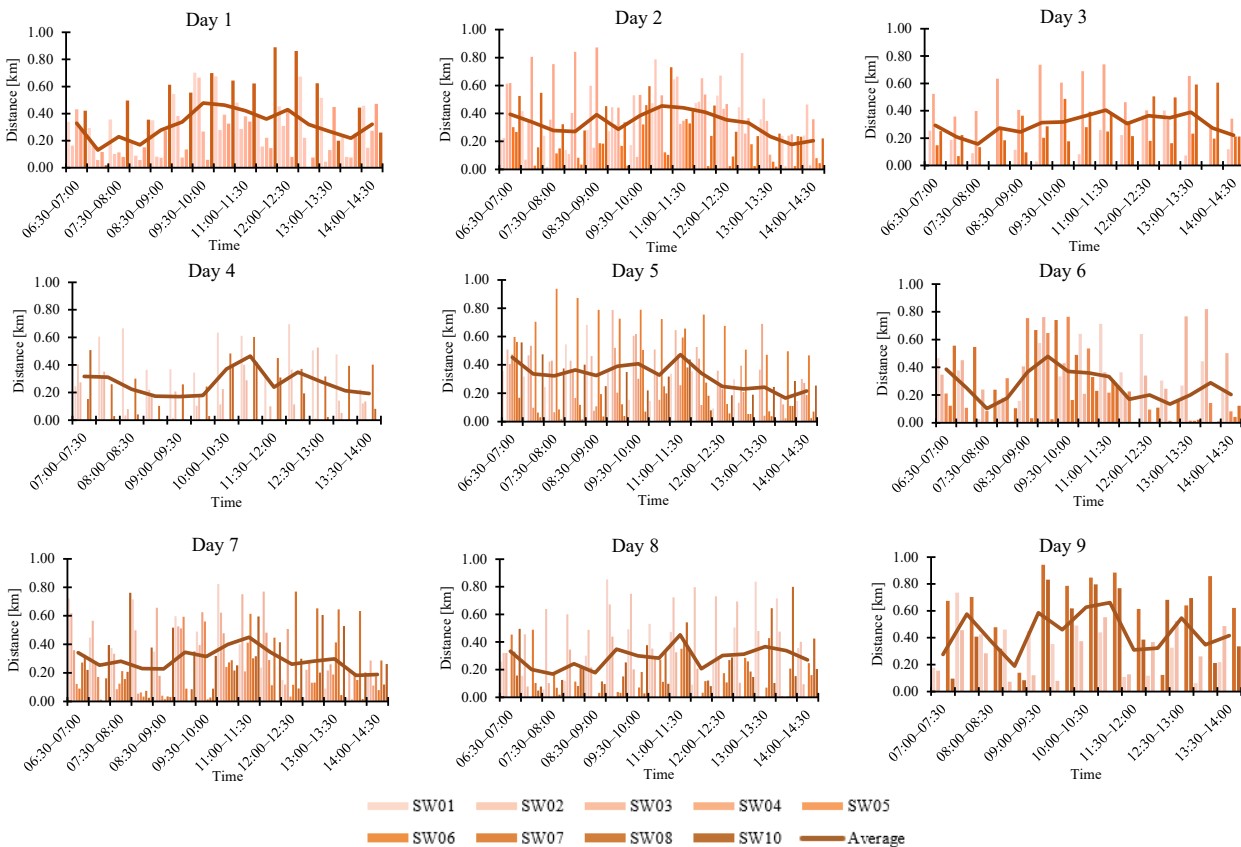

**Figure 3.** Travelled Distances per 30-min interval for each SW on each data collection days.

The data from the smartwatches have been further summarised in Table 6. The time interval with the longest average distance travelled is right after the lunch break at 11:00–11:30 with 0.66 km, closely followed by the time interval right before the break, 10:00–10:30, where the average distance was 0.63 km. This is probably due to the lunchroom being placed far from where the carpenters carried out their tasks, cf. Figure 1. The time interval with the shortest average distance travelled, which was 0.10 km, is at 07:30–08:00. The mean, m, varies from 0.23 km (at 14:00–14:30) to 0.46 km (at 11:00–11:30), and the standard deviation, σ, has a range between 0.05 km (at 06:30–07:00 and 14:00–14:30) and 0.14 km (at 07:30–08:00). Similar analyses could be carried out for each individual day of the data collection.

**Table 6.** Average distance [km] per 30 min of considering all days of data collection.

| Time | Maximum [km] | Minimum [km] | Mean, m [km] | St. dev., σ [km] |
|---|---|---|---|---|
| **06:30–07:00** | 0.45 | 0.29 | 0.36 | **0.05** |
| **07:00–07:30** | 0.34 | 0.13 | 0.26 | 0.07 |
| **07:30–08:00** | 0.58 | **0.10** | 0.27 | **0.14** |
| **08:00–08:30** | 0.38 | 0.17 | 0.26 | 0.07 |
| **08:30–09:00** | 0.39 | 0.17 | 0.26 | 0.08 |
| **09:00–09:30** | 0.59 | 0.17 | 0.36 | 0.12 |
| **09:30–10:00** | 0.48 | 0.18 | 0.36 | 0.09 |
| **10:00–10:30** | 0.63 | 0.28 | 0.41 | 0.10 |
| **11:00–11:30** | **0.66** | 0.33 | **0.46** | 0.09 |
| **11:30–12:00** | 0.41 | 0.17 | 0.30 | 0.08 |
| **12:00–12:30** | 0.43 | 0.20 | 0.31 | 0.07 |
| **12:30–13:00** | 0.55 | 0.14 | 0.31 | 0.11 |
| **13:00–13:30** | 0.39 | 0.20 | 0.28 | 0.07 |
| **13:30–14:00** | 0.42 | 0.17 | 0.25 | 0.09 |
| **14:00–14:30** | 0.32 | 0.19 | **0.23** | 0.05 |

### 4.1.2. Cumulative Travelled Distance

In addition to analysing the distribution of travelled distance along the workdays, the collected data can be depicted as the cumulative distance travelled, as shown in Figure 4. The total travelled distance (TTD) for all 54 activities recorded by the smartwatches can be seen in Figure 4. Cumulative travelled distances per 30-min interval for each SW on each of the data collection days.

Table 7, the shortest distance travelled in one day was 1.07 km (SW07 on Day 6), and the longest distance was 10.10 km (SW06 on Day 5). Both the shortest and the longest average travelled distance in one day occurs on a Friday, where the workday is one hour shorter than the rest of the weekdays: 3.48 km on Day 4 and 5.69 km on Day 9. The remaining seven days, the average varies less than 1 km, namely between 4.03 km (Day 8) and 4.96 km (Day 2).

**Table 7.** The total travelled distance (TTD) recorded for each smartwatch each day of data collection.

| | Day 1 | Day 2 | Day 3 | Day 4 | Day 5 | Day 6 | Day 7 | Day 8 | Day 9 |
|---|---|---|---|---|---|---|---|---|---|
| **SW01** | - | 5.15 | - | 6.08 | 4.07 | - | - | - | - |
| **SW02** | 6.66 | 6.42 | 2.20 | 2.25 | - | 5.70 | 5.63 | 6.00 | - |
| **SW03** | 3.18 | 5.00 | - | 2.71 | 6.69 | - | - | 8.58 | - |
| **SW04** | 3.00 | 7.82 | 7.58 | - | 5.93 | 6.28 | 6.80 | 3.03 | 4.36 |
| **SW05** | 3.25 | - | - | - | 2.63 | - | 4.43 | - | 3.86 |
| **SW06** | - | - | - | - | **10.10** | 4.22 | 3.34 | - | - |
| **SW07** | - | 2.06 | - | - | 3.21 | **1.07** | 2.36 | 2.77 | - |
| **SW08** | - | 2.68 | 3.51 | 4.34 | 1.46 | 4.26 | 4.59 | 2.63 | - |
| **SW09** | - | 5.57 | 4.62 | 2.04 | - | - | 1.93 | 4.07 | 8.31 |
| **SW10** | 7.71 | - | - | - | 4.62 | 2.63 | 6.10 | 3.05 | 6.23 |
| **Average** | 4.76 | 4.96 | 4.48 | **3.48** | 4.84 | 4.03 | 4.40 | 4.30 | **5.69** |



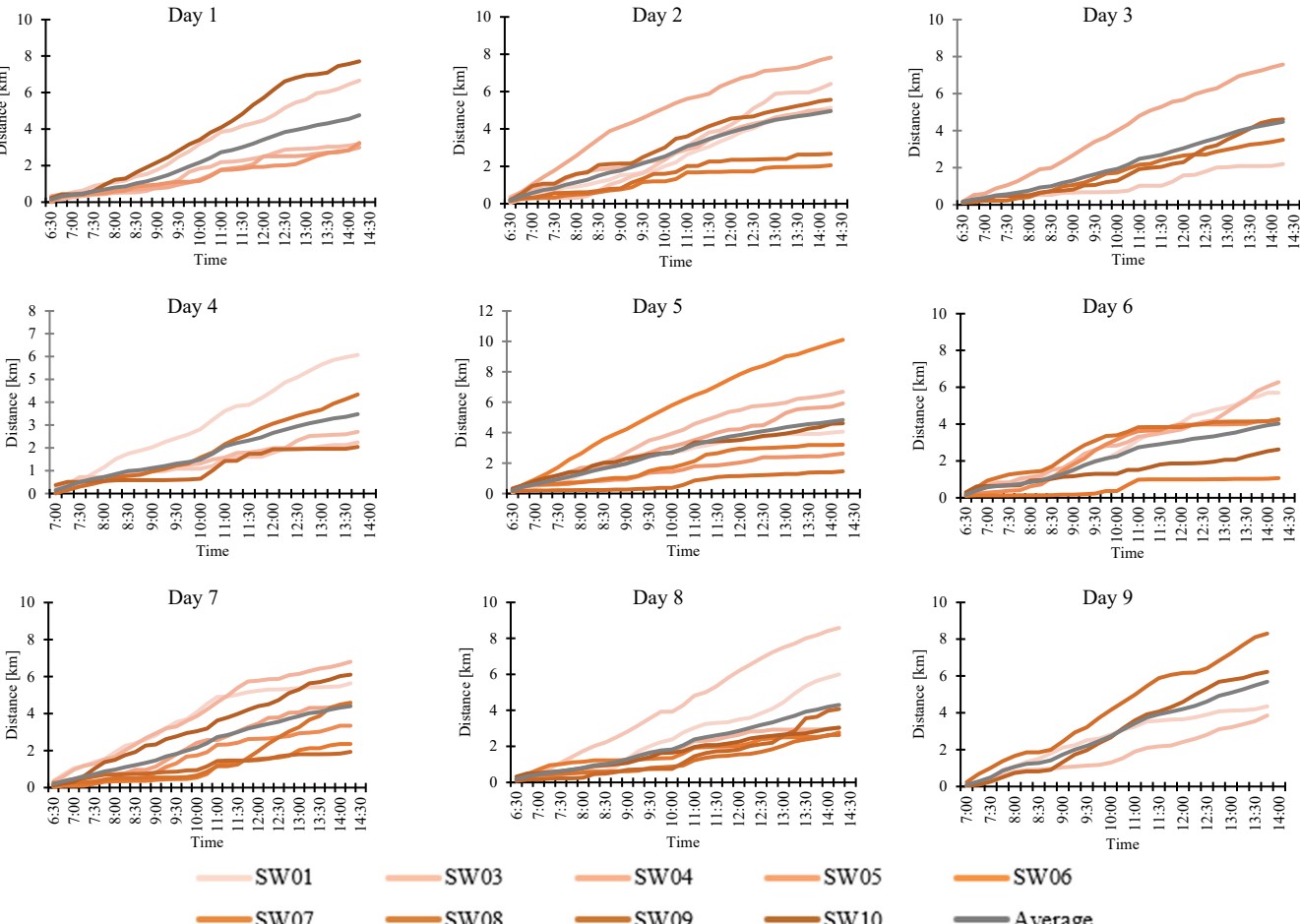

**Figure 4.** Cumulative travelled distances per 30-min interval for each SW on each of the data collection days.

### 4.1.3. Average Travelled Distance [km]

The averages from all nine days of data collection shown in Figure 4 are compiled in Figure 5a. The mean, m, of the nine averages, which vary as described in the section above, is 4.55 km. Figure 5b illustrates the evolvement of the standard deviation, σ, for each of the averages along the day. The standard deviation, σ, varies from 1.71 km on Day 4 to 2.72 km on Day 5, a difference of 45.6%. Further quantitative measures regarding the TTD on the nine days of data collection are listed in Table 8. The longest TTD in one day varies from 6.08 km on Day 4, which was a Friday, to 10.10 km on Day 5. The shortest TTD in one day varies from 1.07 km on Day 6 to 3.86 km on Day 9, which was also a Friday.

**Table 8.** Statistical measures regarding the TTD [km] considering all days of data collection.

|  | Maximum [km] | Minimum [km] | Mean, m [km] | St. dev., σ [km] |
|---|---|---|---|---|
| **Day 1** | 7.71 | 3.00 | 4.76 | 2.25 |
| **Day 2** | 7.82 | 2.06 | 4.96 | 2.01 |
| **Day 3** | 7.58 | 2.20 | 4.48 | 2.29 |
| **Day 4** | 6.08 | 2.04 | **3.48** | **1.71** |
| **Day 5** | **10.10** | 1.46 | 4.84 | **2.72** |
| **Day 6** | 6.28 | **1.07** | 4.03 | 1.93 |
| **Day 7** | 6.80 | 1.93 | 4.40 | 1.76 |
| **Day 8** | 8.58 | 2.63 | 4.30 | 2.22 |
| **Day 9** | 8.31 | 3.86 | **5.69** | 2.02 |

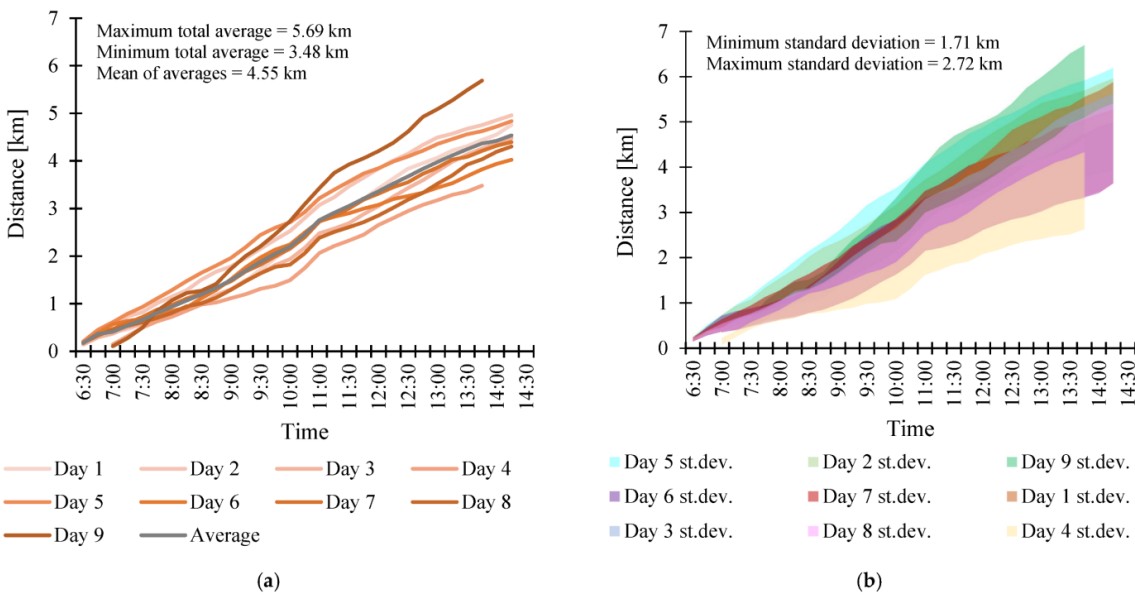

**Figure 5.** (**a**) Cumulative average total distance from the 9 days of data collection; (**b**) Standard deviation of each day.

### 4.2. Workers' Locations on the Job Site

The location of all 63,145 data points collected along the nine days of data collection are shown in Figure 6. The different grey colours indicate each smartwatch, that is, each participating worker of the carpenter trade. It is clear from the charts that the carpenters spent the majority of their time working in the north-western corner of the construction site, i.e., in and around blocks 55, 57, 59, 62, 64, 66, and 68 (see block numbers in Figure 1). On Day 4 and 7, some workers also spent time in blocks 53 and 51.

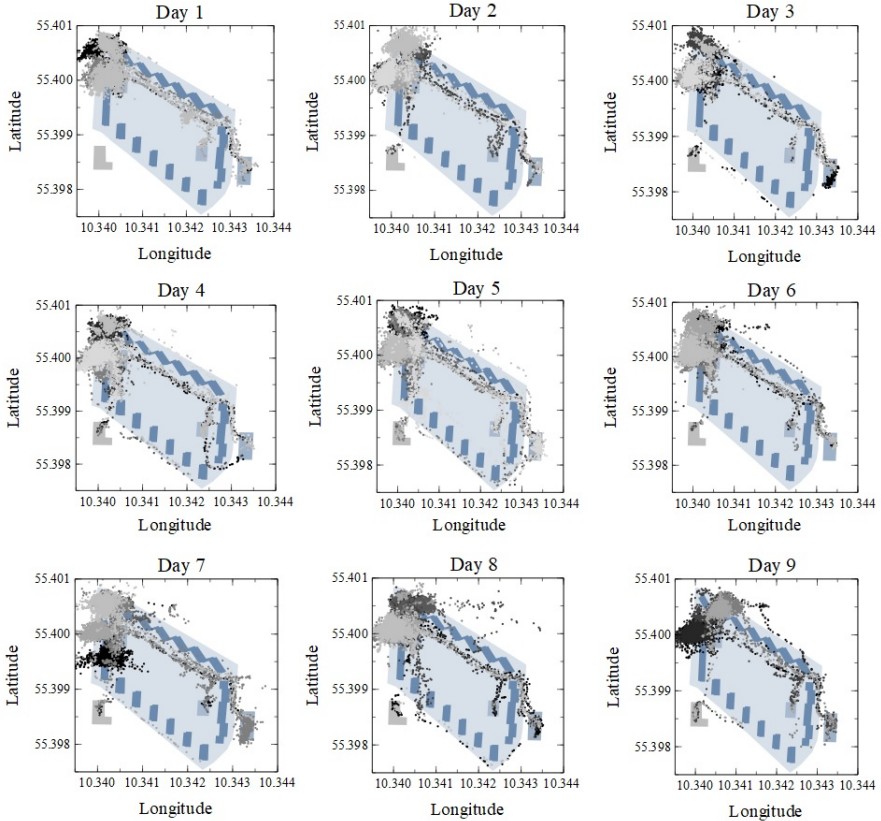

**Figure 6.** Workers' locations on the job site on each of the data collection days.

Additionally, the charts show the most frequently used path for traveling as a straight line along the northern buildings from the building blocks in the northwest, through the gap between block 42 and 18 and down to the offices and break rooms in the most eastern part of the site. The first part of this path is also used to reach the warehouse placed inside the rhomboid shaped yard between the buildings. On all days except Day 1, some of the carpenters visited the supermarket south of the main working area on their way to or from the break room. The data points indicate that more time was spent in the break room on Day 5, 7, 8, and 9 than on the remaining days.

The distribution of data points in Figure 6 shows that several data points were identified far away from the job site. This can, among other reasons, indicate GPS errors when workers conduct activities inside the buildings. This implies that the workers' locations may lack accuracy in detail when zooming in on each geographical point. However, the general distribution can still provide sufficient information for further analysis.

*4.3. Carpenters' Perception*

The authors collected workers' perceptions about the use of smartwatches during their working hours by applying a questionnaire. The answers to the questionnaire are illustrated in Figure 7. Overall, the answers were positive to a large degree. The first question concerned the information presented to the research project participants. Three out of the eight respondents 'strongly agreed' that they had received instructions about the project, and the remaining five respondents 'agreed'. The other three questions in the questionnaire were focused on the smartwatches. Six out of eight respondents 'strongly disagreed' that the watch prolonged their daily tasks or interfered with their work, and two 'disagreed' with these statements. To the last question regarding the carpenters' attitude towards wearing a watch again for future studies, four 'strongly agreed' and three 'agreed' that they would not mind this. One respondent stood out, as he 'strongly disagreed' to wear a watch again. However, this was likely a misunderstanding of the question, because this respondent added a comment to his answers saying; "It was fine wearing the watch".

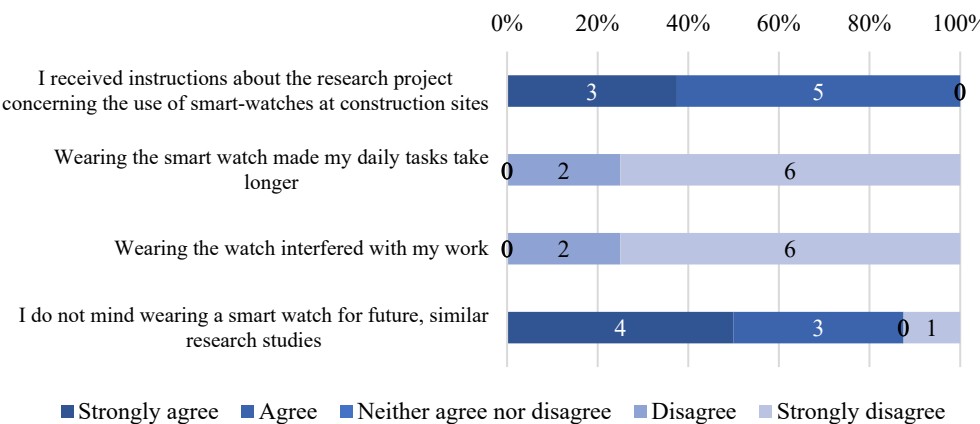

**Figure 7.** The carpenters' perception of wearing a smartwatch while performing their work tasks (*n* = 8).

## 5. Guidelines for the Adoption of Smartwatches to Track Workers

The practical contribution of this study consists of proposing a set of guidelines for future application of smartwatches to collect workers' travelled distances and locations under the umbrella of five categories of lessons learned. The guidelines focus on how researchers and construction practitioners can adopt smartwatches to gather information. Thus, the lessons learned summarise practical recommendations for: (1) selecting participants; (2) smartwatch set up process; (3) data collection; (4) data cleaning; and (5) data analysis. In this section, the authors present some analyses based on the data of the case study to justify and exemplify each lesson learned. For each guideline there is a description, the limitation in the present study, the justification for the adoption of the guideline, and an example of adopting the guideline using the data from the present study.

### 5.1. Guideline 1: Adopt a Stratified Sampling Approach for Selecting the Workers Involved according to Their Tasks Conducted

- Guideline description: During the first step–selecting participants, the authors considered a stratified sampling the most suitable approach because knowing workers' roles can provide helpful information for the analysis phase (e.g., workers in charge of installing windows; workers in charge of installing drywall). Stratified sampling is a method of sampling that involves dividing a population into sub-groups, called strata, based on shared characteristics [47]. Thus, one way to get a better assessment of the workers' duties during the process would be to stratify the sample by type of task conducted.

- Limitation: This paper adopted a random sampling approach for the selection of the participants. As previously described, in this study, all watches were handed out randomly to ensure anonymity, and thus it was not possible to associate the TTD and workers' locations according to their tasks.

- Justification: A justification for the adoption of stratified random sampling can be explained based on the analysis of TTD of this study. It was observed that the foreman spent significantly more time walking than the other carpenters due to the different nature of his tasks. The foreman attended meetings and discussed with management in the office containers and then walked to the different teams of carpenters on site to deliver information and instructions. For this reason, including the foreman in the study when measuring travelled distance could have a distorting impact on the results.

- Exemplifying the justification: To show the impact of including data from the foreman in the analyses, the data from the present study was cleaned of data from the assumed foreman on each day of data collection. Since the foreman was observed walking significantly more than the other participants in the study, in this example, he is assumed to be the one with the longest distance recorded each day. An example from Day 5 of data collection, including and excluding the assumed foreman, respectively, is depicted in Figure 8. The overall shape of the line showing the average distance does not change when excluding the foreman. The average and median distances decrease from 4.84 km to 4.09 km and from 4.35 km to 4.07 km, respectively, but the most significant change is in the standard deviation, σ, which is reduced from 2.72 km to 1.83 km.

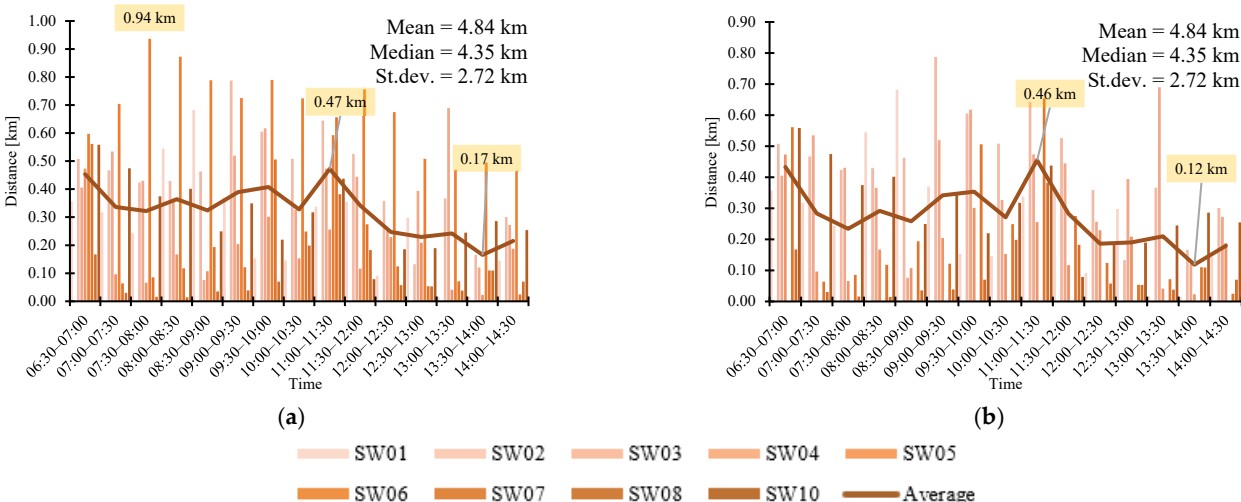

**Figure 8.** TTD per 30-min interval of Day 5, (**a**) Including the assumed foreman in the analysis; (**b**) Excluding the assumed foreman from the analysis.

Assuming the foreman to be the person walking most in one day means eliminating one of the extremes in the data set, which consequently leads to a reduction in the standard

deviation, σ, as shown in Table 9. The reduction varies from 7.77% on Day 6 to 47.16% on Day 3. The significant size of the reduction in standard deviation, 28.19% on average, signifies a considerable difference in travelled distance by the assumed foreman and the rest of the carpenters. Thus, the foreman's tasks and the main duty of each worker should be collected during the first step. Thus, a stratified random sample would have provided a better representation of the TTD according to the role of each worker.

**Table 9.** Change in statistical measures for TTD when excluding the assumed foreman.

| | Description | Maximum [km] | Minimum [km] | Mean, m [km] | St. dev., σ [km] |
|---|---|---|---|---|---|
| **Day 1** | All | 7.71 | 3.00 | 4.76 | 2.25 |
| | Without foreman | 6.66 | 3.00 | 4.02 | 1.76 |
| | Delta | −13.62% | - | −15.55% | −21.78% |
| **Day 2** | All | 7.82 | 2.06 | 4.96 | 2.01 |
| | Without foreman | 6.42 | 2.06 | 4.48 | 1.72 |
| | Delta | −17.90% | - | −9.68% | −14.43% |
| **Day 3** | All | 7.58 | 2.20 | 4.48 | 2.29 |
| | Without foreman | 4.62 | 2.20 | 3.45 | 1.21 |
| | Delta | −39.05% | - | −22.99% | −47.16% |
| **Day 4** | All | 6.08 | 2.04 | 3.48 | 1.71 |
| | Without foreman | 4.34 | 2.04 | 2.83 | 1.04 |
| | Delta | −28.62% | - | −18.68% | −39.18% |
| **Day 5** | All | 10.10 | 1.46 | 4.84 | 2.72 |
| | Without foreman | 6.69 | 1.46 | 4.09 | 1.83 |
| | Delta | −33.86% | - | −15.50% | −32.72% |
| **Day 6** | All | 6.28 | 1.07 | 4.03 | 1.93 |
| | Without foreman | 5.70 | 1.07 | 3.58 | 1.78 |
| | Delta | −9:24% | - | −11.17% | −7.77% |
| **Day 7** | All | 6.80 | 1.93 | 4.40 | 1.76 |
| | Without foreman | 6.10 | 1.93 | 4.05 | 1.58 |
| | Delta | −10.29% | - | −7.95% | −10.23% |
| **Day 8** | All | 8.58 | 2.63 | 4.30 | 2.22 |
| | Without foreman | 6.00 | 2.63 | 3.59 | 1.28 |
| | Delta | −30.07% | - | −16.51% | −42.34% |
| **Day 9** | All | 8.31 | 3.86 | 5.69 | 2.02 |
| | Without foreman | 6.23 | 3.86 | 4.82 | 1.25 |
| | Delta | −25.03% | - | −15.29% | −38.12% |
| **Average change (delta)** | | −23.08% | | −13.33% | −28.19% |
| **Total averages** | All | 5.69 | 3.48 | 4.55 | 0.62 |
| | Without foreman | 4.82 | 2.83 | 3.88 | 0.59 |
| | Delta | −15.29% | −18.68% | −14.73% | −5.14% |

*5.2. Guideline 2: Set Up the Smartwatches Considering Workers' Physical Features*

- Guideline description: During the second step–set up smartwatches, as much as possible, the authors suggest collecting participants' features (e.g., height, weight, gender and age) to add this information to the smartwatch profile, as this will influence the results.
- Limitation: This study adopted average data for a Danish male instead of the personal features of each worker to maintain the anonymity of the smartwatch user.
- Justification: Personal features have an impact on the accuracy of the recorded training data of smartwatches. Height impacts the stride length used to calculate the number of steps taken, and age affects the heart rate zones used to calculate the number of calories burned.
- Exemplifying the justification: The average male walking step length is 75 cm. However, the stride length varied from 63.5 cm for a male of 152 cm to 82.3 cm for a male of 198 cm.

*5.3. Guideline 3: Carefully Consider the Job Site Location for Delivering and Collecting the Smartwatch to Workers*

- Guideline description: During the third step–data collection process, the place to deliver and collect the smartwacthes should be always the same at the job site.
- Limitation: The delivery and collection points for the smartwatches were not consistent, as previously described. This inconsistency had an impact on the recorded total travelled distance.
- Justification: All watches should be delivered and collected at the same place (e.g., the offices or the material storage workspace) to include all travelling necessary for the carpenters to perform their work.
- Exemplifying the justification: The TTDs previously presented in Figure 4 are presented again in Table 10, considering the hypothetical scenario of having delivered and collected the smartwatches at the same place. To understand how the delivery point could have impacted the findings, the distance between Warehouse 2 and the office room (300 m) was included in the cases where the delivered and collected points were different. With this information, the impact of not delivering and collecting the smartwatches by the changing rooms can be measured as the change in TTD. The change in TTD varies between a 4.31% increase and a 35.93% increase.

**Table 10.** Change in TTD when including the distance from the office to the material storage area on site.

| | Delivered | Collected | TTD [km] | TTD if at Office [km] | Δ TTD | | Delivered | Collected | TTD [km] | TTD if at Office [km] | Δ TTD |
|---|---|---|---|---|---|---|---|---|---|---|---|
| **D1-SW02** | Site | Office | 6.66 | 6.96 | 4.31% | **D6-SW02** | Site | Site | 5.70 | 6.30 | 9.52% |
| **D1-SW03** | Site | Site | 3.18 | 3.78 | 15.87% | **D6-SW04** | Site | Site | 6.28 | 6.88 | 8.72% |
| **D1-SW04** | Office | Site | 3.00 | 3.30 | 9.10% | **D6-SW06** | Site | Site | 4.22 | 4.82 | 12.45% |
| **D1-SW05** | Site | Office | 3.25 | 3.55 | 8.45% | **D6-SW07** | Site | Site | 1.07 | 1.67 | 35.93% |
| **D1-SW10** | Halfway | Site | 7.71 | 8.16 | 5.51% | **D6-SW08** | Site | Site | 4.26 | 4.86 | 12.35% |
| **D2-SW01** | Site | Site | 5.15 | 5.75 | 10.44% | **D6-SW10** | Site | Site | 2.63 | 3.23 | 18.58% |
| **D2-SW02** | Site | Site | 6.42 | 7.02 | 8.55% | **D7-SW02** | Office | Site | 5.63 | 5.93 | 5.06% |
| **D2-SW03** | Site | Site | 5.00 | 5.60 | 10.71% | **D7-SW04** | Office | Site | 6.80 | 7.10 | 4.23% |
| **D2-SW04** | Site | Site | 7.82 | 8.42 | 7.12% | **D7-SW05** | Site | Site | 4.43 | 5.03 | 11.93% |
| **D2-SW07** | Site | Site | 2.06 | 2.66 | 22.53% | **D7-SW06** | Site | Site | 3.34 | 3.94 | 15.23% |
| **D2-SW08** | Site | Site | 2.68 | 3.28 | 18.31% | **D7-SW07** | Site | Site | 2.36 | 2.96 | 20.27% |
| **D2-SW09** | Site | Site | 5.57 | 6.17 | 9.73% | **D7-SW08** | Office | Site | 4.59 | 4.89 | 6.13% |
| **D3-SW02** | Site | Site | 2.20 | 2.80 | 21.43% | **D7-SW09** | Site | Site | 1.93 | 2.53 | 23.72% |
| **D3-SW04** | Site | Site | 7.58 | 8.18 | 7.33% | **D7-SW10** | Halfway | Site | 6.10 | 6.55 | 6.87% |
| **D3-SW08** | Site | Site | 3.51 | 4.11 | 14.60% | **D8-SW02** | Site | Site | 6.00 | 6.60 | 9.09% |
| **D3-SW09** | Site | Site | 4.62 | 5.22 | 11.49% | **D8-SW03** | Site | Site | 8.58 | 9.18 | 6.54% |
| **D4-SW01** | Site | Site | 6.08 | 6.68 | 8.98% | **D8-SW04** | Office | Site | 3.03 | 3.33 | 9.01% |
| **D4-SW02** | Office | Site | 2.25 | 2.55 | 11.76% | **D8-SW07** | Site | Site | 2.77 | 3.37 | 17.80% |
| **D4-SW03** | Site | Site | 2.71 | 3.31 | 18.13% | **D8-SW08** | Site | Site | 2.63 | 3.23 | 18.58% |
| **D4-SW08** | Site | Office | 4.34 | 4.64 | 6.47% | **D8-SW09** | Site | Site | 4.07 | 4.67 | 12.85% |
| **D4-SW09** | Office | Site | 2.04 | 2.34 | 12.82% | **D8-SW10** | Site | Site | 3.05 | 3.65 | 16.44% |
| **D5-SW01** | Site | Site | 4.07 | 4.67 | 12.85% | **D9-SW04** | Site | Site | 4.36 | 4.96 | 12.10% |
| **D5-SW03** | Site | Site | 6.69 | 7.28 | 8.24% | **D9-SW05** | Site | Office | 3.86 | 4.16 | 7.21% |
| **D5-SW04** | Site | Site | 5.93 | 6.53 | 9.19% | **D9-SW09** | Site | Site | 8.31 | 8.91 | 6.73% |
| **D5-SW05** | Site | Site | 2.63 | 3.23 | 18.58% | **D9-SW10** | Site | Site | 6.23 | 6.83 | 8.78% |
| **D5-SW06** | Site | Site | 10.10 | 10.70 | 5.61% | | | | | | |
| **D5-SW07** | Site | Site | 3.21 | 3.81 | 15.75% | | | | | | |
| **D5-SW08** | Site | Site | 1.46 | 2.06 | 29.13% | Distance from office to material storage area on site = 300 m. | | | | | |
| **D5-SW10** | Site | Site | 4.62 | 5.22 | 11.49% | | | | | | |

*5.4. Guideline 4: Establish Assumptions for the Data Cleaning Process Regarding Construction Project Features and the Study's Goal*

- Guideline description: During the fourth step–data cleaning process, the authors suggest determining the necessary assumptions for each step to clean the data according to the construction project features. After deciding the best assumptions and steps, researchers should formalize the process for future studies.
- Limitation: The authors of this study adopted three assumptions during the data cleaning process: (1) removing short activities; (2) removing data outside the working hours; and (3) removing data according to the speed of the workers' path. According to the analysis, the third assumption was the only one that did not help clean the data.
- Justification: Establishing and formalizing the premises can be used as a roadmap to follow. Researchers will work inconsistently without formalizing the assumptions adopted according to the construction project features.
- Exemplifying the justification: If the study's goal in adopting smartwatches is to understand how the job site layout influences how much time workers travel, break times should be considered during the analysis. However, if the interest consists of understanding where workers spent their time during the working hours, the break times should be excluded from the study. The following sub-sections present how each of these assumptions could influence the findings.

5.4.1. Removing Activities with a Shorter Duration Than the Workday

The first data cleaning assumption aimed to remove data that did not fit in the dataset regarding the duration of the activities. When the distance travelled by each worker from each smartwatch are compared, the length of the activities should be as close as possible to the working hours length (e.g., eight-hour duration ±30 min). For example, if the interest is to analyse data regarding cumulative distance, but the dataset includes shorter activities (e.g., five-hour duration activities), the researcher must decide to remove or keep these observations. The shorter activities can be considered outliers. So, the researchers need to examine whether the outlier affects the results of the analysis.

In this way, the authors of this paper consider that the first assumption adopted during the cleaning process that reduces the sample size of the study from 64 to 54 activities (see Section 3.3.5) provided a more realistic sample and minimised distraction from the primary target (e.g., to calculate the average cumulative distance by workers during their working hours).

5.4.2. Removing Data Outside Working Hours

The second assumption during the data cleaning aimed to remove data from outside the working hours (i.e., the break times and before and after work). As the scope of this research is limited to the hours of the workdays of the carpenter trade on the renovation project, all data outside of the timeframe for the work hours was removed. For future projects, if the research goal is to understand how the job site logistics influence the travelled distance in a broader way, the dataset should include all necessary walking activities that might take place before and after the paid workhours. Figure 9 provides an example of a visual representation of the GPS coordinates recorded of Day 5 before and after removing the break times from one smartwatch.

In the present study, removing data from the lunch break did not have a significant impact on the results, as can be seen in Table 11. The maximum travelled distance average is reduced by 2.98% or less, and the change in minimum TTD average is between 1.00% and 13.01%. The mean changes with between 0.93% and 3.18%. For six of the days (Day 2–5, 7, and 9), removing data from the break results in an increased standard deviation. For Day 1 and Day 6, the standard deviation decreases, and for Day 8, there is no change in the standard deviation. None of these changes are significant, as the largest change in standard deviation is 3.11% on Day 1.

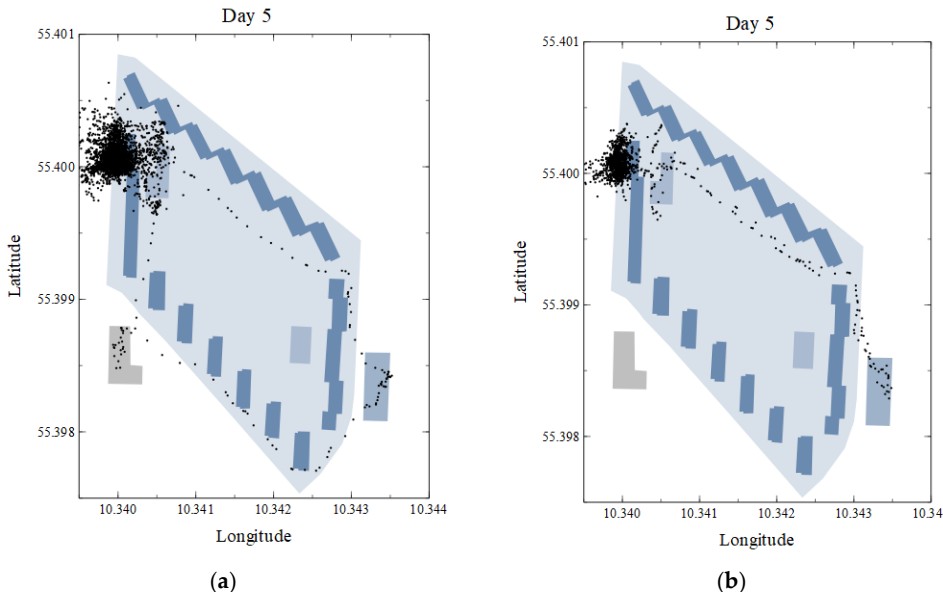

**Figure 9.** Example of job site locations of one single worker on Day 5; (**a**) before removing the break times; (**b**) after removing the break times.

**Table 11.** Change in statistical measures for TTD when including data from the lunch break.

| | Description | Maximum [km] | Minimum [km] | Mean, m [km] | St. dev., σ [km] |
|---|---|---|---|---|---|
| **Day 1** | Without data from break | 7.71 | 3.00 | 4.76 | 2.25 |
| | Including data from break | 7.94 | 3.03 | 4.89 | 2.32 |
| | Delta | −2.98% | −1.00% | −2.73% | −3.11% |
| **Day 2** | Without data from break | 7.82 | 2.06 | 4.96 | 2.01 |
| | Including data from break | 7.96 | 2.30 | 5.09 | 1.98 |
| | Delta | −1.79% | −11.65% | −2.62% | 1.49% |
| **Day 3** | Without data from break | 7.58 | 2.2 | 4.48 | 2.29 |
| | Including data from break | 7.6 | 2.31 | 4.55 | 2.24 |
| | Delta | −0.26% | −5.00% | −1.56% | 2.18% |
| **Day 4** | Without data from break | 6.08 | 2.04 | 3.48 | 1.71 |
| | Including data from break | 6.09 | 2.15 | 3.57 | 1.68 |
| | Delta | −0.16% | −5.39% | −2.59% | 1.75% |
| **Day 5** | Without data from break | 10.1 | 1.46 | 4.84 | 2.72 |
| | Including data from break | 10.25 | 1.65 | 4.97 | 2.69 |
| | Delta | −1.49% | −13.01% | −2.69% | 1.10% |
| **Day 6** | Without data from break | 6.28 | 1.07 | 4.03 | 1.93 |
| | Including data from break | 6.46 | 1.09 | 4.13 | 1.97 |
| | Delta | −2.87% | −1.87% | −2.48% | −2.07% |
| **Day 7** | Without data from break | 6.8 | 1.93 | 4.4 | 1.76 |
| | Including data from break | 6.91 | 2.08 | 4.54 | 1.72 |
| | Delta | −1.62% | −7.77% | −3.18% | 2.27% |
| **Day 8** | Without data from break | 8.58 | 2.63 | 4.3 | 2.22 |
| | Including data from break | 8.64 | 2.72 | 4.34 | 2.22 |
| | Delta | −0.70% | −3.42% | −0.93% | 0.00% |
| **Day 9** | Without data from break | 8.31 | 3.86 | 5.69 | 2.02 |
| | Including data from break | 8.38 | 3.95 | 5.79 | 1.99 |
| | Delta | −0.84% | −2.33% | −1.76% | 1.49% |
| **Total averages** | Without data from break | 5.69 | 3.48 | 4.55 | 0.622 |
| | Including data from break | 5.79 | 3.57 | 4.65 | 0.64 |
| | Delta | −1.76% | −2.59% | −2.20% | −2.89% |

### 5.4.3. Removing Data according to Workers' Speed

The last assumption aimed to clean the collected data from possible GPS errors and data not relevant to this study. An assumption regarding walking speed of workers was applied as described earlier in Section 3.3.5. Figure 10 gives a visual representation of an example of the GPS coordinates recorded during Day 5 for one of the smartwatches before and after the cleaning process using the third assumption. From Figure 10 it can be concluded that this kind of data cleaning has not removed GPS errors completely since there are still some datapoints left outside the job site, where the worker could not have been. Moreover, possible correct datapoints from inside the job site have been removed, which could affect the analyses of the data. For this reason, the authors of this paper suggest not using this assumption in future studies.

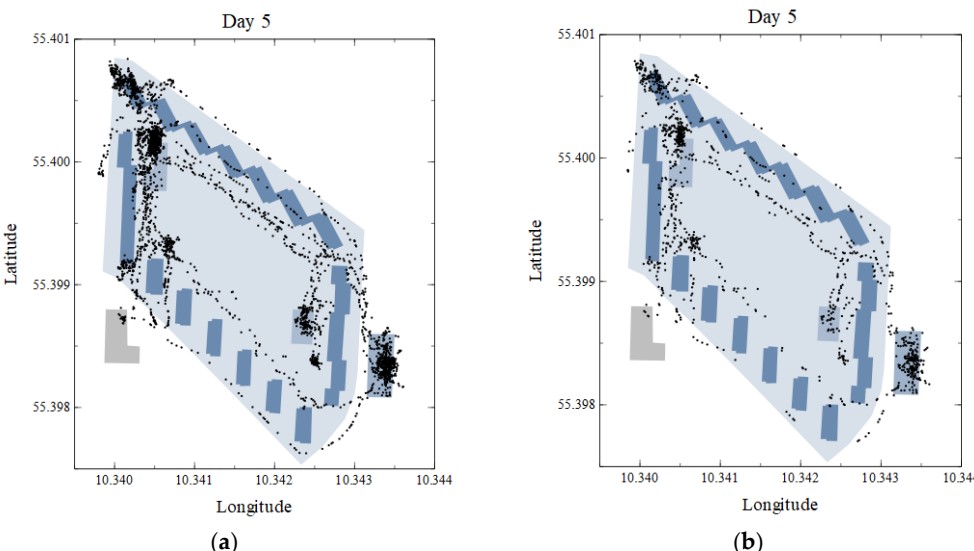

**Figure 10.** Example of job site locations of one single worker on Day 5; (**a**) before removing points regarding speed assumption; (**b**) after removing points regarding speed assumption.

Removing speed below the threshold of 0.5 m/s and removing erroneous data points due to high velocity (>1.48 m/s) resulted in an overall reduction in the number of data points from 194,549 to 63,145, a reduction of 67.54%. This information is shown in Table 12, together with information on each of the 54 usable, recorded activities regarding the number of data points and TTD before and after the cleaning for velocity. The decrease in TTD is substantial, ranging between 46.16% and 69.09%. On average, the total travelled distance is more than halved when applying the speed thresholds; more precisely, it is decreased by 55.40%. This assumption could have impacted the results. However, as this study mainly focuses on presenting the possible analysis that can be conducted with the data gathered rather than providing statistical analysis, the authors considered it reasonable to main the assumption to discuss its impacts.

**Table 12.** Variation in TTD when cleaning data according to speed.

|  | Number of Points Before | Number of Points After | Before [km] | After [km] | Δ [km] | Δ [%] |  | Number of Points Before | Number of Points After | Before [km] | After [km] | Δ [km] | Δ [Δ [%] |
|---|---|---|---|---|---|---|---|---|---|---|---|---|---|
| **D1-SW02** | 4465 | 1842 | 12.37 | 6.66 | 5.71 | −46.18 | **D6-SW02** | 4184 | 1500 | 13.84 | 5.70 | 8.14 | −58.82 |
| **D1-SW03** | 3375 | 862 | 8.35 | 3.18 | 5.17 | −61.91 | **D6-SW04** | 4322 | 1558 | 15.33 | 6.28 | 9.06 | −59.07 |
| **D1-SW04** | 2964 | 776 | 6.33 | 3.00 | 3.33 | −52.67 | **D6-SW06** | 3715 | 1155 | 12.60 | 4.22 | 8.38 | −66.49 |
| **D1-SW05** | 3645 | 897 | 7.05 | 3.25 | 3.80 | −53.90 | **D6-SW07** | 2024 | 211 | 2.66 | 1.07 | 1.60 | −59.93 |
| **D1-SW10** | 3546 | 1630 | 16.82 | 7.71 | 9.11 | −54.17 | **D6-SW08** | 4060 | 1255 | 9.09 | 4.26 | 4.83 | −53.11 |

**Table 12.** *Cont.*

| | Number of Points Before | Number of Points After | Before [km] | After [km] | Δ [km] | Δ [%] | | Number of Points Before | Number of Points After | Before [km] | After [km] | Δ [km] | Δ [Δ %] |
|---|---|---|---|---|---|---|---|---|---|---|---|---|---|
| **D2-SW01** | 4081 | 1375 | 10.25 | 5.15 | 5.10 | −49.78 | **D6-SW10** | 3461 | 678 | 6.14 | 2.63 | 3.52 | −57.26 |
| **D2-SW02** | 4172 | 1612 | 14.19 | 6.42 | 7.77 | −54.76 | **D7-SW02** | 3802 | 1418 | 13.41 | 5.63 | 7.77 | −57.99 |
| **D2-SW03** | 4534 | 1290 | 9.60 | 5.00 | 4.60 | −47.89 | **D7-SW04** | 4410 | 1743 | 12.96 | 6.80 | 6.16 | −47.55 |
| **D2-SW04** | 4399 | 1824 | 17.32 | 7.82 | 9.49 | −54.82 | **D7-SW05** | 3776 | 1139 | 11.50 | 4.43 | 7.08 | −61.52 |
| **D2-SW07** | 2928 | 519 | 6.42 | 2.06 | 4.36 | −67.86 | **D7-SW06** | 3221 | 870 | 7.93 | 3.34 | 4.58 | −57.82 |
| **D2-SW08** | 2919 | 616 | 8.67 | 2.68 | 5.99 | −69.13 | **D7-SW07** | 3211 | 644 | 5.92 | 2.36 | 3.56 | −60.19 |
| **D2-SW09** | 3894 | 1457 | 12.56 | 5.57 | 6.99 | −55.68 | **D7-SW08** | 2640 | 1196 | 11.54 | 4.59 | 6.95 | −60.22 |
| **D3-SW02** | 2587 | 532 | 5.87 | 2.20 | 3.66 | −62.42 | **D7-SW09** | 2539 | 484 | 4.38 | 1.93 | 2.45 | −55.94 |
| **D3-SW04** | 4823 | 1972 | 14.67 | 7.58 | 7.09 | −48.35 | **D7-SW10** | 3851 | 1421 | 15.47 | 6.10 | 9.37 | −60.57 |
| **D3-SW08** | 3628 | 955 | 8.71 | 3.51 | 5.19 | −59.65 | **D8-SW02** | 4122 | 1602 | 12.23 | 6.00 | 6.23 | −50.95 |
| **D3-SW09** | 4336 | 1242 | 9.29 | 4.62 | 4.64 | −49.94 | **D8-SW03** | 5247 | 2383 | 17.93 | 8.58 | 9.35 | −52.16 |
| **D4-SW01** | 4233 | 1712 | 12.11 | 6.08 | 6.04 | −49.84 | **D8-SW04** | 3691 | 877 | 7.32 | 3.03 | 4.29 | −58.60 |
| **D4-SW02** | 1221 | 487 | 5.42 | 2.25 | 3.17 | −58.53 | **D8-SW07** | 3025 | 650 | 6.22 | 2.77 | 3.45 | −55.48 |
| **D4-SW03** | 3407 | 770 | 7.37 | 2.71 | 4.67 | −63.31 | **D8-SW08** | 2982 | 725 | 7.22 | 2.63 | 4.58 | −63.49 |
| **D4-SW08** | 3566 | 1158 | 8.77 | 4.34 | 4.43 | −50.50 | **D8-SW09** | 3082 | 961 | 7.80 | 4.07 | 3.73 | −47.79 |
| **D4-SW09** | 2324 | 434 | 4.36 | 2.04 | 2.32 | −53.20 | **D8-SW10** | 3790 | 822 | 7.64 | 3.05 | 4.59 | −60.12 |
| **D5-SW01** | 3920 | 1070 | 9.49 | 4.07 | 5.42 | −57.11 | **D9-SW04** | 3281 | 1119 | 8.68 | 4.36 | 4.32 | −49.79 |
| **D5-SW03** | 4821 | 1834 | 13.65 | 6.69 | 6.96 | −51.02 | **D9-SW05** | 3484 | 1097 | 8.28 | 3.86 | 4.42 | −53.37 |
| **D5-SW04** | 3385 | 1367 | 12.76 | 5.93 | 6.84 | −53.56 | **D9-SW09** | 4266 | 2141 | 17.77 | 8.31 | 9.46 | −53.25 |
| **D5-SW05** | 2874 | 640 | 7.35 | 2.63 | 4.73 | −64.26 | **D9-SW10** | 3814 | 1627 | 13.76 | 6.23 | 7.53 | −54.73 |
| **D5-SW06** | 5467 | 2629 | 20.01 | 10.10 | 9.91 | −49.52 | **Total** | 194,549 | 63,145 | 548.77 | 244.77 | | |
| **D5-SW07** | 2930 | 862 | 8.47 | 3.21 | 5.25 | −62.05 | **Average** | 3603 | 1169 | 10.16 | 4.55 | 5.61 | −55.2 |
| **D5-SW08** | 2282 | 339 | 3.58 | 1.46 | 2.11 | −59.12 | | | | | | | |
| **D5-SW10** | 3823 | 1166 | 9.34 | 4.62 | 5.71 | −46.18 | | | | | | | |

*5.5. Guideline 5: Use Individual Participant Data in the Analysis according to Each Participant's Characteristics and Role*

- Guideline description: During the fifth step–data analysis, the adoption of individual participant data (IPD) can provide a more appropriate analysis according to each participant's characteristics and role. This approach requires the ability to identify which participant uses each smartwatch each day, preferably by delivering the same smartwatch to the same participant every day during the study period. For this, the product identification number of the smartwatch could be associated with the name or the role of each worker. This way, when the study ends, the names can be deleted to ensure the anonymity of the workers without the loss of data.

- Limitation: This study presented the results using an aggregated data approach based on the whole body of the collected data. The data aggregation process classified the data collected using the smartwatches according to the day of data collection and the smartwatch used (e.g., D1-SW01). The random delivery of each smartwatch to protect the anonymity of the workers prevented the adoption of an IPD approach.

- Justification: The major advantage of an IPD analysis compared to the adopted aggregated data approach is that it allows detailed participant-level exploration of effectiveness in relation to individual characteristics such as role (e.g., foreman, crew member) or construction task conducted (e.g., workers in charge of installing windows; workers in charge of installing drywall).

- Exemplifying the justification: The same example used in the Guideline 1 can be used for justifying the adoption of IPD. An IPD would have provided a better representation of the TTD and the locations on the job site according to the role of each worker.

## 6. Discussion

This study presents several contributions in the discipline in which it is applied. The study explores the adoption of location-based sensor technology from a smartwatch to understand how and where construction workers spend their working time. The present study focuses on, mainly, workers' travelled distance; and workers' location on the job site, as a new approach for tracking workers. Applying smartwatch to track workers lacks previous research attention.

The first managerial contribution consists of identifying workers's travelled distance. The results presented in Section 4.1 illustrate approaches to understanding total travelled distance, cumulative travelled distance, and average travelled distance. These approaches can provide several useful insights to construction site management for further analysis.

Firstly, the representation of the total travelled distance grouped in 30-min intervals will support the identification of which hour of the day the workers are most likely to travel. This kind of analysis can be useful for site managers to check workers' travel patterns along the day, and plan and adjust resources accordingly. The total travelled distance index can also be used to compare site layouts between construction sites. By using a formula like "travelled distances/construction site area ($m^2$)" or "travelled distances/construction site volume ($m^3$)", managers can get numbers which allow comparisons between projects. In both indexes, a high number might indicate low worker efficiency, which might be due to workspace conflicts or inefficient construction site layout planning. Therefore, the total travelled distance index, in combination with other construction site size measurements, can be used as an indicator for workspace management.

Another interesting insight can be drawn from the cumulative distance travelled by workers to understand workers' performance. As the higher distance travelled, the less time is spent in the same location. The cumulative distance travelled by each worker can be an indicator of actual time spent in value-adding activities. For instance, if in a construction project, the travelled distance on the job site of different trades presents a high distance, this indicator could raise alarms to notice potential logistic issues. Therefore, further studies can use this indicator to calculate the presence in work zones and, consequently, the performance of workers.

Lastly, the average travelled distance analysis will support understanding the nature of different construction activities. It is well known that construction processes using prefabricated and off-site methods present a smaller share of time on direct work on the job site. Consequently, the implementation of prefabrication components on the job site will increase, among other factors, the time spent in preparation and transportation activities. Thus, the average distance measurement can be useful to understand the standard deviation among the distance travelled by workers regarding the nature of the construction processes that they are involved in.

The second managerial contribution is the identification of workers' outdoor locations. The results shown in Section 4.2 indicated how this information could be useful for construction management.

Firstly, the density of points throughout the different days can provide the identification of where the carpenter trade's activities were executed during the period of the observations. Using this 2D representation to see where construction workers on the job site are located can be used during planning meetings with contractors to see the location of possible problems, thus forming the starting point for discussion. This way, objective data presented in visual form can be a joint base for learning. By discussing the variation of worker's locations during the week, each trade can explain to the others what causes changes of locations in their workflow, and this way, all trades obtain a greater overview of the renovation process. So, the illustrations can allow trade supervisors and managers to solve minor problems and coordinate their work schedules, thus preventing minor issues from growing.

A second analysis regarding the density of points of workers' locations consists of developing heat maps for logistic purposes. The distributions of points along the job site

presented in this study were limited to visualise the location rather than to understand the density of points on each location. Construction managers can use the aggregated visualisation of workers' locations to discover job site usage patterns and make data-informed optimisations to improve workers' distribution and avoid congestion areas. In addition, heat maps with hourly intervals using the timestamp of each point can be useful to identify and manage high density locations along the workday.

A third analysis can be regarding dividing the job site into workspaces. This analysis will support identifying workers' presence in different workspaces. For this, the distribution of points can be classified into different workspaces categories according to the goal of the specific study (e.g., production workspace, storage workspace, preparation workspace, and transportation workspace). To conduct this analysis, the geographical coordinates of each workspace can be used to define workspace areas. Then, using a programming language, each geographical point collected with the smartwatch can be loaded, analysed, plotted, and reported according to the workspace taxonomy adopted. With this approach, the distribution of workers' presence in workspaces can be used to understand workers' performance based on where workers spend their time on construction sites. Thus, workers' presence could be used to correlate with value-adding time.

The workspace analysis could also include the distribution of workers' locations with hourly intervals. As presented in the Section 4.1.1. regarding the total travel distance, the workers' locations on the job site can be discussed regarding the hour of the day. The workers' location movements throughout the working hours could provide further insights regarding workspace presence in different hours.

Moreover, the literature review of previous studies that adopted technologies for measuring construction workers' performance and the discussion above allowed the authors of this paper to establish the hypothesis that workers' efficiency can be measured indirectly by their travelled distances and locations.

*6.1. Future Steps*

The present paper presents the first learning cycle of an ongoing research project which aims to automate monitoring of workers' activities. The case study conducted in the first learning cycle served as the starting point for defining the DSR problem. The problem came from the literature review with the identification of the knowledge gap about how to use location-based technologies, specifically wristband sensors, for tracking workers and from the first case study.

Based on the results of the first case study, a practical solution with a theoretical contribution will be developed as the artifact of this DSR. The main goal of this phase was to understand which types of information smartwatches can collect and how this data can be employed for measuring workers' efficiency. The current study identified a set of potential uses of travelled distances and locations of workers.

In the next steps of this research, the artifact will be developed, tested, and evaluated in Cycles 2, 3 and 4 in Case B, C and D, respectively. The present authors will conduct three new case studies on other construction projects, applying the lessons learned from this study. The authors will conduct the next studies in projects with the same characteristics as Case A. The next three case studies will be selected considering the following features: (1) renovation projects of multi-story buildings; (2) use of scaffolding for conducting external activities; and (3) limited use of equipment for conducting transportation activities. In addition, the authors will choose the renovation projects considering that the carpenter trade conducts most of the renovation activities. The reasons for limiting the study application to the carpenter trade are: (1) to be able to compare the data among construction projects with the same characteristics conducted by the same kind of workers; and (2) the tangible nature of the activities performed by carpenters.

During the construction phase in Cycle 2 in Case B, the authors will develop the artifact. A different carpenter trade will be selected for the use of the smartwatches during their working hours. In the second case study, the Work Sampling (WS) technique will be

applied to identify how workers spent their time. For this purpose, the random observations collected during the WS application will be classified into value-adding activities and non-value-adding activities. Consequently, this technique will allow the authors to measure workers' efficiency based on the time spent on value-adding activities. Data gathered from the smartwatches will be analysed to determine where the activities were carried out. As in the present study, the smartwatches will provide two indexes: workers' travelled distance and workers' location. Then, the following analyses will be conducted: (1) the relationship between workers' locations and workers' travelled distances; (2) the relationship between workers' locations and time spent on value-adding activities; (3) the relationship between travelled distance and worker's efficiency.

The authors will test and improve the artifact in Cycle 3 in Case C. In this third case study, the WS technique will be applied, as in Case B, to identify the time spent by workers in value-adding and non-value-adding work categories, classified as productive, contributory and non-contributory. Still, in this case, the data gathered from smartwatches will be analysed to determine the time spent in different workspaces. So, in Case C, instead of using the data provided by the smartwatches to measure workers' travelled distances and location, the data will be used to identify the distribution of time in different work areas characterized as productive workspace; contributory workspace; and non-contributory workspace. In this cycle, the authors will conduct the following analysis: (1) the relationship between workers' travelled distance and time spent in workspaces; and (2) the relationship between workers' efficiency and time spent in each workspace.

In Cycle 4, the last version of the artifact will be evaluated. The evaluation phase will concern the theoretical contribution that emerges from the research process. The main output of the DSR will be a method for understanding workers' efficiency indirectly by measuring workers' travelled distances and workers' locations collected by smartwatches. During Cycle 4, several criteria will be chosen to assess the evaluation of the method developed. Lastly, the utility and usefulness of the method will be evaluated in a fourth case study, named in this research as Case D. The feedback of this evaluation will lead to a refinement of the proposed method.

*6.2. Limitations*

In addition to multiple benefits, the present study presented several limitations. One of the main limitations of this exploratory case study may be described as the study preparation step, previously discussed during the guideline proposition. Other main limitations were: (1) the random sampling approach during the selection of the participants did not allow to associate the results with the nature of the activity conducted by the user; (2) the physical features of each individual worker were not set up on the devices, and the lack of this information could have influenced the value of the distance travelled; (3) the inconsistency of the delivery and collected places of the smartwatches to workers impacted on the recorded total travelled distance; (4) the assumptions adopted during the data cleaning process consisting of removing data according to the speed of the workers' path impacted the findings; and (5) the adoption of aggregated data approach based on the whole body of the collected data prevented the association between travelled distance and workers' positions according to the workers' tasks.

Another limitation is that, at this time, the proposed approach is only able to be used for outdoor activities due to the technological constraints of the GPS used by the smartwatches. However, it was observed during the case study development that some workers conducted some tasks inside the buildings under renovation. This might have impacted the results obtained in the total travelled distance and workers' locations. Besides, the error range of the total travelled distance can be larger than focusing the workers' locations themselves. Considering that each smartwatch gathered around 3600 data points during 8 working hours, and a few of the data points were wrongly located. The workers' locations are still good enough to provide a general idea of the workers' distribution of time on the job site. However, as the total travelled distance is the accumulative distance

measured between two consecutive points, if one of those points has a wrong position due to GNSS errors, the impact on the result will be bigger. Hence, in future studies, the authors will choose workers that exclusively conduct their tasks from outside the buildings.

Lastly, in the first case study, the data points collected from smartwatches were analysed to exclusively identify travelled distance and workers' locations. Such analyses aim at the calculation of workers' efficiency. However, smartwatches can collect other sorts of information such as Heart Rate and Skin Temperature. If more types of data are collected, more indexes can be calculated. Consequently, workers' efficiency could be measured as the combination of several indicators.

*6.3. Recommendations for Future Research*

This research raised topics to be examined in greater depth in future research efforts. These are summarized below.

This study presents the first step of a research project that aims to propose a method for adopting smartwatches to measure workers' efficiency based on workers' travelled distances and locations. Although the method is being built based on the results of case studies on building renovation projects, future studies should adopt the smartwatches in different construction projects to evaluate their utility in other projects where workers spend most of their time outdoor. Some examples of other construction projects can be: (1) infrastructure projects, such as highways, streets, and roads; (2) rail projects; and (3) airport projects.

Another topic for future investigation is to further explore the relationships between workers' travelled distance and workers' locations and other indexes collected by smartwatches, such as Heart Rate and Skin Temperature Indexes, as previously described. This research aimed to measure workers' efficiency based exclusively on the location-based sensors in the smartwatches. Further research can explore the indexes collected by other sensors embedded in the devices.

An interesting approach for future studies will be comparing the workers' locations collected by smartwatches and by other digital technologies. Smartphones can present several advantages to identifying the device's location, as those devices combine GNSS information with other sources of data, such as the internet connection and mobile data connection.

**7. Conclusions**

This paper presents the first step of a novel approach to measuring workers' efficiency automatically. In the last decades, the existing literature regarding the adoption of tools to automate monitoring workers' activities indicated growing attention to using sensors, more precisely wristband-type activity trackers. The literature pointed out the difficulty of adopting more sophisticated approaches by practitioners due to the necessity of extensive data training and extended data period analysis. For this reason, this research adopted smartwatches.

Based on the finding of previous studies that adopted smartwatches, the main features and advantages of using a smartwatch as a research tool are: (1) they represent a non-invasive and non-subjective measurement of user physiological parameters in a mobile environment [27]; (2) they consist of flexible approaches to evaluate the mental state of users [25]; (3) they provide a collection of geolocation information via GNSS to understand user mobility patterns [15]; they do not interfere in the mobility of the user [48]; (4) they are easy to access due to the commercial availability of different models and brands in the market [14]; (5) they represent a low implementation and low maintenance cost [33]; (6) they are energy efficient because their low-power consumption as much of the time the device is in ultra-low power standby-mode [49] The smartwatch used in this study was the Garmin Forerunner 45; however, other devices with similar features can be used in future studies.

The research strategy adopted was Design Science Research (DSR), considering the real problem and the necessity to build a solution for it. The solution is represented by the artifact of this research project, which consists of a method for understanding workers' efficiency indirectly by measuring workers' travelled distances and workers' locations collected by smartwatches. The method is being designed and evaluated over four learning cycles. This paper exclusively focuses on presenting the results of the first learning cycle. Cycle 1 was a relevant cycle for understanding which types of information smartwatches can collect and how this data can be employed for measuring workers' efficiency. The outcome analysis of Cycle 1, based on results from Case Study A, contributed to answering the two research questions. Those were addressed as follows:

- How can smartwatches be adopted to facilitate understanding workers' travelled distances and job site location?

The exploratory case study conducted as Case A allowed the authors to understand the possible analyses with the adoption of smartwatches to track workers. The findings of the study lead the authors to provide a generic set of five recommendations for collecting workers' travelled distance and workers' location using smartwatches. The guidelines are: (1) adopt a stratified sampling approach for selecting the workers involved according to their tasks conducted; (2) set up the smartwatches considering workers' physical features; (3) carefully consider the job site location for delivering the smartwatch to workers; (4) establish assumptions for the data cleaning process regarding construction project features and the study's goal; and (5) use individual participant data in the analysis according to each participant's characteristics and role.

- How can the data gathered using smartwatches be helpful in measuring workers' efficiency?

Based on this question, the second contribution focuses on identifying the possible analyses that can be conducted from the workers' travelled distances and the workers' locations. The analysis of the travelled distance by workers during working hours allowed the authors to identify possible uses of this data: (1) the identification of which hour of the day the workers are most likely to travel; (2) the measurement of workers' performance (i.e., time spent in value-adding activities) based on the amount of travel along the day; and (3) the understanding of the nature of different construction activities using as a basis the workers' time spent travelling. The distribution of workers' locations within the job site can also provide interesting analyses: (1) the identification of where the workers' activities are being conducted to see where potential problems are and identify possible congested areas on the job site; (2) the measurement of workers' presence in different workspaces; (3); the distribution of workers' locations with hour interval using the timestamp of geographical location; and (4) the development of heat maps to understand the density of points on each location.

The main novelty of this research lies in an innovative way of obtaining workers' travelled distance and job site location using smartwatches. While previous studies that adopted wristworn sensors to facilitate automatic monitoring of workers' movements, those approaches have been generally tested in simulated scenarios and, when tested in job sites, have been limited to a reduced number of previously labelled activities. While other methods for identifying workers' locations can be very cumbersome to apply by practitioners on the job site, the use of an affordable smartwatch has a huge potential of applicability for outdoor activities. Smartwatches offer researchers the benefit of data collection in an objective and non-invasive way. The questionnaire answers of the workers involved in this study revealed that they did not feel the use of a smartwatch interfered with their work.

**Author Contributions:** Conceptualization, methodology, formal analysis, resources, writing—original draft preparation, writing—review and editing, C.T.P., S.S. and S.W.; funding acquisition, S.W. All authors have read and agreed to the published version of the manuscript.

**Funding:** This research was funded by the Independent Research Fund Denmark grant number 0217-00020B.

**Institutional Review Board Statement:** Not applicable.

**Informed Consent Statement:** Informed consent was obtained from all subjects involved in the study.

**Acknowledgments:** The authors would like to thank the construction company for opening their job site doors.

**Conflicts of Interest:** The authors declare no conflict of interest.

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
