# Peer review of "Five Guidelines for Adopting Smartwatches in Construction: A Novel Approach for Understanding Workers’ Efficiency Based on Travelled Distances and Locations"

_sustainability, doi:10.3390/su14148875_

Round 1
Reviewer 1 Report
Dear Authors
Thank you for submitting a comprehensive case study paper exploring the use of smartwatches for essentially what I would consider determining and improving productivity. I have read through it many times and must say that from the start I had trouble following and finding a red line. The paper overall is good, novel, and interesting. I recommend a more sound structure that merits a scientific approach. A clear research question must be presented by which you would follow with the methodology that is based on the thus far unaddressed gaps in knowledge presented in the literature.
Maybe consider making the title more catchy: for instance
The Adoption of Smartwatches in construction: A novel aspect of workers efficiency with the focus of travel distances and locations.
I recommend separate sections. Theoretical contributions – your paper focuses on practical contributions almost entirely. Limitations should be separately outlined. Develop the discussion. And most importantly develop the conclusions more comprehensively.
I am missing the productivity aspect of it, an important area of research dealing with digitalization as the main result of the use of advanced technologies. Please consider the following paper for reference. 10.1007/s11846-022-00543-7
Author Response
The authors thank the reviewers for their comments to improve not only the quality of the content but also the style of this paper. The modifications are highlighted in blue and the moved text is in green in the revised version for quick access. The authors addressed your suggestions as presented in the attached file.

Reviewer 2 Report
The paper presents an exploratory case study of using smartwatches for tracking employees. The paper is well-written and in my opinion, suitable for publication after minor revision.
Strong points of the paper:
- a real-life case study based on a carpenter trade building renovation project in Odense in Denmark,
- the set of five guidelines for the usage of smartwatches in data collecting projects.
Suggestions for improvements:
- As the paper is quite long, summarizing the related works in Table 1 is a very good idea. I also suggest shortening the description of the Literature review part.
- The research design steps could be illustrated using a diagram/chart to be easier to comprehend.
- The guidelines could be also highlighted in the abstract.
- The guidelines could be summarized in a tabular way with the additional descriptions of the required equipment, constraints or other elements which need to be taken into account, limitations, and importance (if it is possible to determine).
I would also consider changing the title to the more appealing one, i.e., instead of "Lessons Learned from", it could be, e.g., "Five guidelines for...", which could increase the interest in the paper.
Minor issues:
- p. 13: lack of the reference to Table 4: "The excluded 446 activities are highlighted in Error! Reference source not found.".
- Figure 5b is not really clear as too much information is presented in the same picture.
Author Response
The authors thank the reviewers for their comments to
improve not only the quality of the content but also
the style of this paper. The modifications are
highlighted in blue and the moved text is in green in
the revised version for quick access. The authors
addressed his suggestions as described in the attached file.
